# Defining classical and quantum chaos through adiabatic transformations

Cedric Lim[a], Kirill Matirko[b], Hyeongjin Kim[c],
Anatoli Polkovnikov[c], and Michael O. Flynn[c,*]

[a] *Department of Physics, University of California, Berkeley, California 94720, USA*

[b] *Department of Physics, HSE University, St. Petersburg 190008, Russia*

[c] *Department of Physics, Boston University, Boston, MA 02215, USA*

December 17, 2024

## Abstract

We propose a formalism which defines chaos in both quantum and classical systems in an equivalent manner by means of *adiabatic transformations*. The complexity of adiabatic transformations which preserve classical time-averaged trajectories (quantum eigenstates) in response to Hamiltonian deformations serves as a measure of chaos. This complexity is quantified by the (properly regularized) fidelity susceptibility. Physically this measure quantifies long time instabilities of physical observables due to small changes in the Hamiltonian of the system. Our exposition clearly showcases the common structures underlying quantum and classical chaos and allows us to distinguish integrable, chaotic but non-thermalizing, and ergodic/mixing regimes. We apply the fidelity susceptibility to a model of two coupled spins and demonstrate that it successfully predicts the universal onset of chaos, both for finite spin $S$ and in the classical limit $S \to \infty$. Interestingly, we find that finite $S$ effects are anomalously large close to integrability.

## Contents

*michaelflynn13@gmail.com

# 1   Introduction

Chaos and ergodicity are two closely related cornerstones of physics. In classical systems chaos is defined through the instability of a particle's trajectories to small variations of either initial conditions or the Hamiltonian. It is widely believed that chaos usually leads to ergodicity or equivalently thermalization, meaning that a system "forgets" its initial conditions except for conserved quantities, such as energy, and approaches an equilibrium state under its own dynamics. A standard, oft-quoted argument states that the time-averaged probability distribution

$$\overline{P}(\boldsymbol{x}, \boldsymbol{p}) = \lim_{T \to \infty} \frac{1}{T} \int P(\boldsymbol{x}, \boldsymbol{p}, t) \tag{1}$$

for any bounded motion is time-independent. Therefore, it satisfies the stationary Liouville equation, $\{H, \overline{P}\} = 0$, where $H \equiv H(\boldsymbol{x}, \boldsymbol{p})$ is the Hamiltonian of the system and $\{\cdots\}$ denotes the Poisson bracket. As such, $\overline{P}(\boldsymbol{x}, \boldsymbol{p})$ can only depend on conserved functions of phase space variables [1]. For systems with a time-independent Hamiltonian and without continuous symmetries, the only conserved quantity is the Hamiltonian itself and this argument suggests that $\overline{P}(\boldsymbol{x}, \boldsymbol{p}) = \overline{P}(H(\boldsymbol{x}, \boldsymbol{p}))$. For any given trajectory with a fixed energy $E$, we then have $\overline{P}(H(\boldsymbol{x}, \boldsymbol{p})) \propto \delta(E - H(\boldsymbol{x}, \boldsymbol{p}))$, which is equivalent to the microcanonical ensemble.

    Chaos appears only indirectly in this argument. It is believed that, in general, motion becomes unstable or chaotic if the number of independent degrees of freedom describing a classical system is larger than the number of independent conservation laws. When energy is the only conserved quantity, we expect that the classical motion of a single particle is both chaotic and ergodic in $d > 1$ spatial dimensions. However, in such few-particle systems, the relation between chaos and ergodicity is more subtle. For example, a famous theorem due to Kolmogorov, Arnold, and Moser (KAM) [2] states that ergodic behavior generally occurs only if the system is sufficiently far from

an integrable point [3]. There is, however, *no* such threshold for the emergence of chaos. Namely, at weak integrability breaking phase space generally becomes mixed, containing regions of regular and chaotic motion. So while, at least pictorially, chaos and ergodicity appear for the same reason (breakdown of extra conservation laws which exist in integrable systems), the precise relation between these two concepts is highly nontrivial.

The definition of chaos in quantum mechanics is even more debated, as there is no clear notion of trajectories or Lyapunov exponents. A single quantum state corresponds not to a point in a phase space but rather to a probability distribution, which is usually stable even classically. For example, the motion of a single Brownian particle can be chaotic, while the probability distribution describing an ensemble of such particles after relatively short times evolves according to the diffusion equation, which has no exponential instabilities.

One attempt to define quantum chaos in analogy with classical systems was through the short time behavior of so-called out of time order correlation functions (OTOC) [4–8]. A particularly well-studied example of OTOCs are correlation functions of the form $|\langle [\hat{A}(t), \hat{B}]^2 \rangle|$, where $\hat{A}$ and $\hat{B}$ are (initially local) operators in the Heisenberg representation and the expectation value is taken with respect to some initial state. In the classical limit the commutator between two operators, up to a factor of $i\hbar$, reduces to the Poisson bracket between corresponding observables evaluated at different times. The square of the commutator thus approaches the square of the Poisson bracket and generally diverges with an exponent that is twice the classical Lyapunov exponent. It was rather quickly realized that OTOCs generally do not exhibit exponential growth in quantum systems with locally bounded Hilbert spaces and thus cannot be used to define quantum chaos [9–11].

Recently, a set of related ideas emerged suggesting that quantum chaos can be detected through the growth of operators in Krylov space [12–16]. This operator spreading is related to the high frequency tail of the corresponding spectral function, which in turn describes the noise and dissipative response of the system. An operator which spreads rapidly has a spectral function with a slowly decaying high frequency tail; chaotic systems exhibit the fastest allowed operator spreading consistent with constraints such as locality, and therefore the slowest possible decay of spectral functions. This suggestion was numerically confirmed in one-dimensional quantum spin chains [17,18]. However, as we will discuss below, operator growth shows very similar asymptotic behavior in both chaotic and integrable models near the classical limit, that is, it does not allow one to clearly distinguish chaotic and integrable systems (see also Ref. [19]).

Perhaps the most accepted definition of quantum chaos is based on the Berry and Berry-Tabor conjectures formulated for billiards [20,21] and later extended by O. Bohigas, M. Giannoni, and C. Schmit (BGS) [22]. The BGS conjecture states that energy eigenstates of quantum chaotic systems are essentially random vectors with eigenvalues described by appropriate random matrix ensembles. This is to be contrasted with generic integrable systems, where according to the Berry-Tabor conjecture the level statistics are approximately Poissonian, such that nearby energy levels are effectively uncorrelated. The BGS conjecture was subsequently generalized independently by J. Deutcsh and M. Srednicki to the eigenstate thermalization hypothesis (ETH) [23,24], which was spectacularly confirmed via numerical experiments [25]. Now ETH is accepted as a standard definition of quantum chaos (see Ref. [26] for a review) and it is usually tested numerically by studying either level spacing distributions [27,28] or the asymptotic behavior of the spectral form factor

$$K(\tau) = \frac{1}{Z} \left| \sum_n \exp[-2\pi i E_n \tau] \right|^2,$$

and a closely related return probability [28]. In systems which obey ETH, $K(\tau)$ shows a characteristic linear growth with time $\tau$ at long times due to level repulsion [28–31]. We stress that despite being accepted as a definition of chaos, both ETH and the BGS

conjecture are really statements about ergodicity or long-time thermalization [32]. Indeed, these conjectures imply that the time averaged density matrix

$$\bar{\rho} = \lim_{T \to \infty} \rho(t)dt = \sum_n \rho_{nn}|n\rangle\langle n|, \tag{2}$$

where $|n\rangle$ are the eigenstates of the Hamiltonian, behaves as a thermal ensemble. This is a direct analogue of the classical definition of ergodicity through the time averaged probability distribution (1). Likewise, ETH can be used to prove various thermodynamic relations like the fluctuation-dissipation relation or Onsager relations for individual eigenstates [26]. While chaos and ergodicity often come together, these concepts are not equivalent and we should be able to distinguish them. Chaos usually implies a lack of predictability encoded in high sensitivity of various observables to weak perturbations. Ergodicity, on the other hand, is related to approaching thermal equilibrium at long times. There are plenty of examples, such as models described by KAM theory, where a system can be chaotic but non-ergodic and conversely.

Let us note that these standard definitions of classical and quantum chaos are at odds with our everyday experience. Typically, a colloquial definition of chaos is based on the so-called "butterfly effect", which states that small perturbations in a system might result in a large, observable change of its later state. For example, a flap of a butterfly's wing in one part of the world can completely change the path of a tornado far away and at later times. However, it is intuitively clear that this effect is not related to Lyapunov instabilities. Indeed, tornadoes associated with turbulent instabilities happen in systems with small viscosity like air, which are weakly interacting and hence have small Lyapunov exponents. If we take a more viscous liquid like water with much larger Lyapunov exponents, then following a flap of the wing (or rather a fin), the liquid will quickly relax to a new local equilibrium and the effect of the flap on observables will be minimal no matter how long we wait. Moreover, collisions between atoms can be quantum mechanical such that trajectories and Lyapunov exponents are not even well-defined. Nevertheless, the long-time instabilities and hence chaos can exist irrespective of this fact. Likewise, this example indicates that our everyday experience suggests that the strongest chaos occurs in systems which are far from thermal equilibrium, i.e., which are close to integrability. Hence, measures based on thermalization cannot be used to define chaos. This is in addition to the fact that eigenstates are not even-well defined in macroscopic systems.

In Ref. [33] a different approach to quantum chaos was developed through the sensitivity of Hamiltonian eigenstates to small perturbations. In classical mechanics the role of eigenstates is played by stationary (time-averaged) trajectories. Mathematically this sensitivity is expressed through the scaling of the fidelity susceptibility, which is also also known as (up to irrelevant multiplicative factors) the quantum Fisher information, the quantum geometric tensor, or the norm of the adiabatic gauge potential. This probe was successfully tested in various systems undergoing crossovers between integrable and ergodic regimes [34–38]. In this way, one can distinguish different dynamical regimes and interestingly, an intermediate regime separating integrable and ergodic domains has been observed. This intermediate regime is maximally chaotic in the sense that the fidelity susceptibility saturates bounds on its growth and diverges with system size more rapidly than in the ergodic regime. We note that this notion of "maximal chaos" is different from others in the literature, such as the saturation of bounds on OTOC growth [39] or in Floquet dual unitary circuits [40]. In our opinion, those models are better termed as "maximally thermalizing", "maximally ergodic", or "maximally mixing", but not "maximally chaotic", for reasons we explain below.

The intermediate regime is physically characterized by anomalously slow, glassy dynamics of observables, reminiscent of Arnold diffusion in classical systems [3]. It was observed numerically in interacting quantum models, with or without disorder [33–35, 38, 41]. A schematic crossover diagram showing the transition from integrable to ergodic behavior in extensive quantum systems which summarizes the findings of these references is shown in Fig. 1. While in the thermodynamic limit in generic interacting systems only the ergodic regime survives, there is always a para-

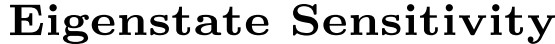

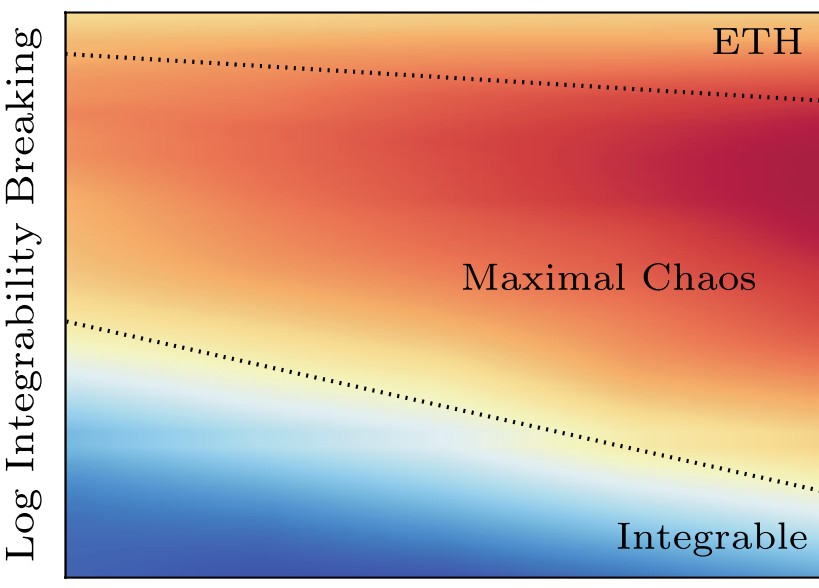

Figure 1: A schematic diagram showing the crossover from integrable to ergodic behavior in quantum spin systems. The horizontal axis stands for the system size and the vertical axis is an integrability breaking perturbation on a log scale. The color temperature indicates the fidelity susceptibility, which is a measure of eigenstate sensitivity, also on a log scale. The integrable and ergodic ETH phases are separated by a broad chaotic but non-ergodic regime characterized by maximal eigenstate sensitivity. The plot is courtesy of M. Pandey and is adopted from the popular summary of Ref. [33].

metrically wide crossover regime. In the thermodynamic limit this regime manifests itself through transient, slow relaxation of autocorrelation functions [41], which are also known to exist in classical systems like the Fermi-Pasta-Ulam model [42]. Let us also point out that there are parallels between this maximally chaotic regime and Hilbert multifractality of eigenstates close to integrability, bearing close analogies to mixed phase space in classical systems [43,44]. The fact that maximal chaos defined in this way occurs at small integrability breaking perturbations agrees with our everyday intuition as we discussed above.

While this way of defining and probing chaos seems to be intrinsically quantum mechanical as it relies on the notion of eigenstates, the main focus of the present work is to show that this is not the case. In fact, the fidelity susceptibility can be used just as well to probe and define chaos in classical systems and allows us to construct a universal probe of chaos, integrability, and ergodicity in both quantum and classical systems. As we discuss below, in classical systems the fidelity susceptibility characterizes the complexity of special canonical transformations which leave invariant stationary probability distributions (equivalently, time-averaged trajectories), which are direct analogues of quantum eigenstates.

Colloquially, the fidelity susceptibility tells us how difficult it is to deform the canonical variables after adding a small perturbation to the Hamiltonian such that the time averaged classical trajectories in the new coordinates remain unaffected by this perturbation. Strictly speaking, in chaotic systems such trajectory-deforming transformations do not exist [45]. However, as we discuss below, one can regulate

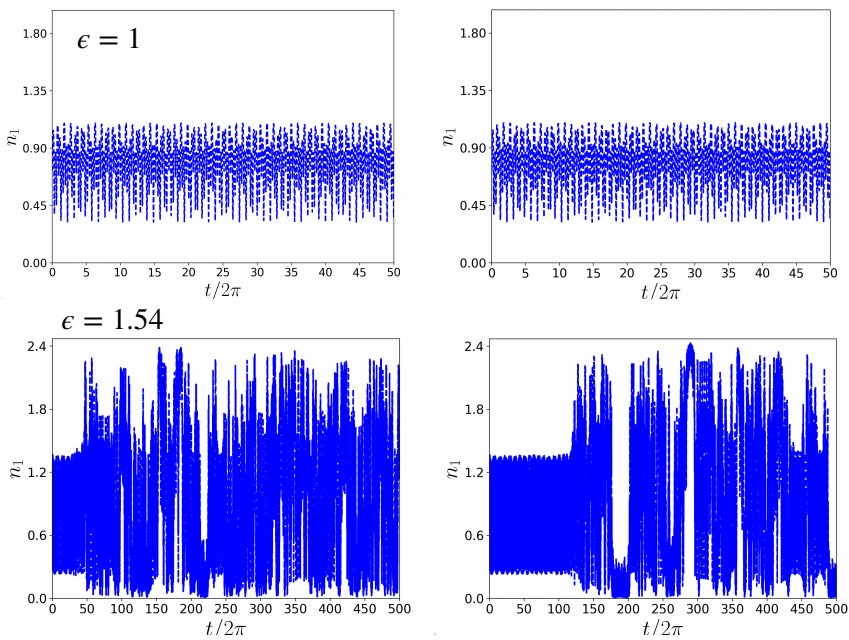

Figure 2: Time dependence of the observable $n_1(t)$ for two 2D classical nonlinear oscillators with frequencies $\omega = 1$ (left) and $\omega = 1.001$ (right) and identical initial conditions (see text for details). Clearly, $n_1(t)$ responds strongly to the shift of $\omega$, and the adiabatic transformation relating these trajectories is very complex. This, in turn, implies that the fidelity susceptibility associated with shifting $\omega$ is large. This setup lies in the "maximal chaos" regime (see Fig. 1 and surrounding discussion), so that in addition to significant variation of individual trajectories, the distribution functions obtained by averaging them over long times also vary strongly.

the generator of such a transformation using a finite-time cutoff and define chaos through the asymptotic behavior of the fidelity susceptibility with this cutoff. The same methodology can be used to analyze large quantum systems, where the energy levels are so dense that distinguishing them is not feasible [41]. The fidelity susceptibility characterizes the long-time behavior of autocorrelation functions (equivalently, low-frequency behavior of spectral functions) and in this way is complementary to the high-frequency spectral data encoded in the short-time behavior of operators. In quantum systems, the fidelity susceptibility can be defined for individual eigenstates and as such has a distribution, which can be nontrivial [34,46]. Likewise in classical systems one can define a separate susceptibility for long-time trajectories originating at different initial conditions. One can anticipate that close to integrability, where phase space is mixed, the distribution of fidelity susceptibilities — like the distribution of Lyapunov exponents — will be very broad. In this paper, we focus on analyzing the phase space-averaged (or in quantum systems, Hilbert space-averaged) fidelity susceptibility, which misses potential coexistence of regular and chaotic regions. However, we will demonstrate that in such a mixed regime we can easily distinguish chaotic and regular phase-space regions by resolving the fidelity susceptibilities over the corresponding trajectories.

To gain a more intuitive understanding of the fidelity approach to chaos let us consider a simple example of a classical particle with unit mass in a two-dimensional nonlinear potential described by the following Hamiltonian:

$$H = \frac{p_1^2}{2} + \frac{p_2^2}{2} + \frac{\omega_1^2 x_1^2 + \omega_2^2 x_2^2}{2} + \frac{\epsilon}{4} x_1^2 x_2^2. \tag{3}$$

As an observable, we choose $n_1 = p_1^2/(2\omega_1) + \omega_1 x_1^2/2$, which represents the adiabatic

invariant of the first oscillator in the absence of the nonlinearity, i.e. at $\epsilon = 0$. In Fig. 2, we plot four different trajectories for this system. Each trajectory corresponds to the same initial conditions: $x_1(0) = 1.2$, $p_1(0) = 0$, $x_2(0) = 0.8$, $p_2(0) = 0$. The two left (right) panels correspond to $\omega_1 = \omega_2 = 1$ ($\omega_1 = 1.001$, $\omega_2 = 1$), respectively. In other words, instead of comparing trajectories with slightly different initial conditions, we consider trajectories with slightly different Hamiltonians. The top (bottom) plots correspond to the nonlinearity $\epsilon = 1$ ($\epsilon = 1.54$). It is visually obvious that the motion in the top (bottom) panels is regular (chaotic). This fact can be quantified by analyzing the scaling of the distance between the two trajectories in time, which is a standard method for measuring chaoticity. Alternatively, one can consider the complexity of the canonical mapping which transforms the trajectories in the right panels to those in the left panels. As we explain in this paper, the fidelity susceptibility, i.e. the measure of this complexity, is encoded in a very different scaling of the long time temporal fluctuations of $n_1(t)$ along regular and chaotic trajectories. Additionally in the bottom (chaotic) panels one can clearly observe large deviations of $n_1(t)$ from the time average, which indicates that the motion of the particle is non-ergodic in the accessible phase space domain. As we will demonstrate below, the scaling of the fidelity susceptibility with a time cutoff allows one to unambiguously differentiate chaotic-ergodic and chaotic non-ergodic trajectories, with the latter being more sensitive to small perturbations and hence more chaotic.

The paper is organized as follows. Sec. 2 discusses how the fidelity susceptibility serves as a probe of chaos and develops the necessary formalism to compute it in both quantum and classical systems. We also demonstrate that operator growth cannot generally distinguish chaotic and integrable systems. In Sec. 3, we introduce a model of two interacting spins which can exhibit both integrable and chaotic dynamics depending on the choice of model parameters. The remainder of the paper is then focused on studying that model both for finite spin $S$ and in the classical limit $S \to \infty$. In particular, we study the spectral function of a particular physical observable in different regimes, which include a highly symmetric integrable model (Sec. 4), an integrable model without any continuous symmetries (Sec. 5), and a chaotic model at different values of the strength of the integrability breaking perturbation (Sec. 6). As we will see, the quantum spectral function converges to the classical result in all cases. Interestingly, close to integrability this convergence is anomalously slow. Moreover, the low-frequency behavior of the spectral function sharply distinguishes between chaotic and integrable cases that lead to qualitatively different behaviors of the fidelity susceptibility, which we can tune by either using different model parameters with whole phase-space averaging, or focusing on a single model parameter but averaging over different regions of phase space. We find in Sec. 6.1 that finite $S$ effects lead to weaker signatures of chaos in quantum systems, which is reminiscent of dynamical localization in a kicked rotor [47]. Using the fidelity approach, we establish that this model is not ergodic (that is, it is always in a mixed phase space regime and does not thermalize for any strength of the integrability breaking parameter) and support this conclusion with other standard measures of ergodicity. In Sec. 6.3, we demonstrate that one can distinguish chaotic and regular trajectories by analyzing the low frequency tails of the auto-correlation functions of observables and/or corresponding fidelity susceptibilities.

# 2 Fidelity Susceptibility and Operator Growth in Quantum and Classical Systems

We use this section to review some formalism and recent results which should prove useful to readers. Sections 2.1 and 2.2 introduce the adiabatic gauge potential and its relationship to the fidelity susceptibility, the primary quantity of interest to this work. In particular, we emphasize its mathematical representation and physical meaning in the classical limit. We also review aspects of operator growth and its connection

to spectral functions in Sec. 2.3, which has recently been the subject of extensive discussion. Experts familiar with these topics may choose to skip to Sec.3, where we introduce the model which is our focus in the remainder of this work.

## 2.1 Adiabatic Gauge Potential

Here we will largely summarize earlier results on the relation of the fidelity susceptibility to adiabatic transformations and related response functions, highlighting similarities and differences between quantum and classical systems. In the former, adiabatic deformations are generated by the adiabatic gauge potential (AGP) $\mathcal{A}_\lambda$:

$$i\hbar\partial_\lambda|n(\lambda)\rangle = \mathcal{A}_\lambda|n(\lambda)\rangle, \tag{4}$$

where $\lambda$ is a coupling in the Hamiltonian and $|n(\lambda)\rangle$ are instantaneous eigenstates of $H(\lambda)$. By differentiating the identity $\langle m(\lambda)|n(\lambda)\rangle = \delta_{mn}$ with respect to $\lambda$ it is easy to see that $\mathcal{A}_\lambda$ is a Hermitian operator. The AGP can be viewed as the generator of unitary transformation $U(\lambda)$ which diagonalize the Hamiltonian, that is, if $|n(\lambda)\rangle = U(\lambda)|n(0)\rangle$ then

$$\mathcal{A}_\lambda = i\hbar(\partial_\lambda U)U^\dagger. \tag{5}$$

In the absence of degeneracies, we can apply stationary perturbation theory to Eq. 4 and find

$$\langle n|\mathcal{A}_\lambda|m\rangle = i\hbar\langle n|\partial_\lambda|m\rangle = i\hbar\frac{\langle n|\partial_\lambda H|m\rangle}{E_m - E_n}. \tag{6}$$

Multiplying both sides of this equation by $E_n - E_m$ we see that the AGP satisfies [48]

$$[G_\lambda, H] = 0, \quad G_\lambda = \partial_\lambda H + \frac{i}{\hbar}[\mathcal{A}_\lambda, H]. \tag{7}$$

Alternatively, one can obtain this equation by requiring that $\tilde{H}(\lambda) = U^\dagger(\lambda)H(\lambda)U(\lambda)$ is represented by a diagonal matrix in a fixed basis $|n(0)\rangle$ for any value of $\lambda$, which implies that $[\tilde{H}(\lambda), \tilde{H}(\lambda + \delta\lambda)] = 0$. Differentiating this equation with respect to $\delta\lambda$ yields Eq. (7). In this form the AGP is well-defined irrespective of degeneracies. However, the AGP is not unique as one can, for example, add to it any operator which commutes with the Hamiltonian. This ambiguity reflects the gauge freedom in the definition of eigenstates, hence the name AGP. Equation (7) is also well defined in the classical limit,

$$\{G_\lambda, H\} = 0, \quad G_\lambda = \partial_\lambda H - \{\mathcal{A}_\lambda, H\}, \tag{8}$$

where $\{\dots\}$ denotes the Poisson bracket.

In integrable systems, the AGP is the generator of special canonical transformations which preserve adiabatic invariants [49]. To see the meaning of the AGP in a generic system, observe that the Hamiltonian deformation $H(\lambda) \to H(\lambda + \delta\lambda)$ is equivalent to shifting the Hamiltonian by a conserved function $H(\lambda) \to H(\lambda) + G_\lambda\,\delta\lambda$ followed by the canonical transformation $\boldsymbol{x}^\lambda \to \boldsymbol{x}^{\lambda+\delta\lambda}$, $\boldsymbol{p}^\lambda \to \boldsymbol{p}^{\lambda+\delta\lambda}$ satisfying

$$\frac{\partial x_j^\lambda}{\partial\lambda} = -\frac{\partial\mathcal{A}_\lambda}{\partial p_j^\lambda}, \quad \frac{\partial p_j^\lambda}{\partial\lambda} = \frac{\partial\mathcal{A}_\lambda}{\partial x_j^\lambda}. \tag{9}$$

Using the chain rule and Eq. (9),

$$H(\boldsymbol{x}^\lambda, \boldsymbol{p}^\lambda, \lambda + \delta\lambda) = H(\boldsymbol{x}^{\lambda+\delta\lambda}, \boldsymbol{p}^{\lambda+\delta\lambda}, \lambda) + G_\lambda(\boldsymbol{x}^\lambda, \boldsymbol{p}^\lambda, \lambda)\,\delta\lambda + O(\delta\lambda^2). \tag{10}$$

Alternatively we can interpret this transformation using the fact that

$$H(\boldsymbol{x}^{\lambda-\delta\lambda}, \boldsymbol{p}^{\lambda-\delta\lambda}, \lambda + \delta\lambda) = H(\boldsymbol{x}^\lambda, \boldsymbol{p}^\lambda, \lambda) + G_\lambda(\boldsymbol{x}^\lambda, \boldsymbol{p}^\lambda, \lambda)\,\delta\lambda + O(\delta\lambda^2), \tag{11}$$

which implies that, up to a conserved operator, an infinitesimal change of the Hamiltonian $H(\lambda) \to H(\lambda + \delta\lambda)$ can be undone (up to an energy shift) by an infinitesimal deformation of canonical variables, $x^\lambda \to x^\lambda - \partial_\lambda x^\lambda\delta\lambda$, $p^\lambda \to p^\lambda - \partial_\lambda p^\lambda\delta\lambda$, where $\partial_\lambda x^\lambda$

and $\partial_\lambda p^\lambda$ are defined by Eq. (9). Therefore, if we view the deformed phase space variables $\boldsymbol{x}^{\lambda-\delta\lambda}$ and $\boldsymbol{p}^{\lambda-\delta\lambda}$ as functions of $\boldsymbol{x} \equiv \boldsymbol{x}^\lambda$ and $\boldsymbol{p} \equiv \boldsymbol{p}^\lambda$, they will evolve according to the Hamiltonian $H(\boldsymbol{x}, \boldsymbol{p}, \lambda) + G_\lambda(\boldsymbol{x}, \boldsymbol{p}, \lambda)\delta\lambda$. We see that when $G_\lambda = 0$, meaning that there are no thermodynamic generalized forces associated with the parameter $\lambda$, the deformed variables under the deformed Hamiltonian simply follow the original trajectories. When $G_\lambda$ is non-zero this canonical transformation generally changes particular time dependent trajectories but leaves invariant their time-averages, which, as we discussed, describe stationary probability distributions. This is understood, in the language of quantum mechanics, by noting that such a commuting perturbation does not generally change the eigenstates. Since the time-averaged density matrix of any pure state is equivalent to a statistical mixture of these eigenstates it also does not change, provided that we start from the same initial state. [1]

From Eq. (6) one can see that the AGP can be represented as [45,50,51]

$$\mathcal{A}_\lambda = \frac{1}{2} \int_{-\infty}^{\infty} dt \, \mathrm{e}^{-\mu|t|}\mathrm{sgn(t)}\partial_\lambda H(t), \tag{12}$$

where

$$\partial_\lambda H(t) = \mathrm{e}^{\frac{i}{\hbar}Ht}\partial_\lambda H \mathrm{e}^{-\frac{i}{\hbar}Ht} \equiv \partial_\lambda H(\boldsymbol{x}(t), \boldsymbol{p}(t))$$

is the Heisenberg representation of the operator $\partial_\lambda H$ in a quantum system. Classically, it is a function $\partial_\lambda H(\boldsymbol{x}, \boldsymbol{p})$, which is evaluated in time-dependent phase space coordinates $\boldsymbol{x}(t), \boldsymbol{p}(t)$, evolved with the Hamiltonian $H$. We have also introduced the cutoff $\mu$, a small positive number, to regularize the integral. With this regularization, the AGP is defined for all systems of interest to us, whether they are integrable, chaotic, quantum, classical, finite or infinite. For quantum (classical) systems the regularized AGP defines unitary (canonical) transformations which preserve $G_\lambda$ up to a time $1/\mu$. In the language of quantum mechanics this statement also means that $G_\lambda$ has suppressed matrix elements between states with energy differences $|E_n - E_m| > \hbar\mu$. To see the equivalence of Eqs. (12) and (6) we can evaluate the matrix elements of Eq. (12) explicitly by performing the time integral:

$$\langle n|\mathcal{A}_\lambda|m\rangle = i\frac{\omega_{nm}}{\omega_{nm}^2 + \mu^2}\langle n|\partial_\lambda H|m\rangle, \quad \omega_{nm} = \frac{E_m - E_n}{\hbar}. \tag{13}$$

Clearly as $\mu \to 0$ this reproduces Eq. (6). Similarly, one can check that the approximately conserved operator defined through the regularized AGP, $G_\lambda = \partial_\lambda H + i[\mathcal{A}_\lambda, H]/\hbar$, is given by the time average of $\partial_\lambda H(t)$,

$$G_\lambda = \frac{\mu}{2} \int_{-\infty}^{\infty} dt \, \mathrm{e}^{-\mu|t|}\partial_\lambda H(t). \tag{14}$$

In the limit $\mu \to 0$, $G_\lambda$ thus represents the so-called Drude weight, or conserved part of $\partial_\lambda H$. The operators $G_\lambda$ can be highly nontrivial; for example, in Ref. [52] they were used to identify previously unknown quasi-local conservation laws in the integrable XXZ chain.

Finally, let us note that Eq. (7) furnishes a variational approach to computing the AGP and the conserved operator $G_\lambda$. Namely, this equation is equivalent to minimizing the action [48]

$$\frac{\delta\mathcal{S}}{\delta\mathcal{A}_\lambda} = 0, \quad \mathcal{S} = \|G_\lambda\|^2 + \mu^2\|\mathcal{A}_\lambda\|^2,$$

$$\|G_\lambda\|^2 = \sum_n \rho_n \langle n|G_\lambda^2|n\rangle \xrightarrow[\hbar\to 0]{} \int d\boldsymbol{x}d\boldsymbol{p}\, P(\boldsymbol{x}, \boldsymbol{p})G_\lambda^2(\boldsymbol{x}, \boldsymbol{p}), \tag{15}$$

where $\|\mathcal{A}_\lambda\|^2$ is defined similarly to $\|G_\lambda\|^2\|$ [2], and $\rho_n$ is an arbitrary stationary (diagonal in the Hamiltonian basis) density matrix, which is replaced in the classical limit

---

[1]There are non-generic situations where the time-averaged distribution can be changed. For example, this is the case if $H$ has a set of degeneracies due to a symmetry which is lifted by $G$.

[2]Strictly speaking only the connected parts of the operators $G_\lambda^2$ and $\mathcal{A}_\lambda^2$ enter the norms (see Eq. (21)). However, this subtlety does not affect the action minimization.

by a stationary probability distribution $P(\boldsymbol{x}, \boldsymbol{p})$. For example, one could choose the Gibbs ensemble

$$\rho_n = \frac{1}{Z} \mathrm{e}^{-\beta E_n}, \quad P(\boldsymbol{x}, \boldsymbol{p}) = \frac{1}{Z} \mathrm{e}^{-\beta H(\boldsymbol{x}, \boldsymbol{p})}, \tag{16}$$

where $Z$ is the usual partition function. In the extreme case of infinite temperature $\beta = 0$ the averaging is carried out uniformly over all quantum eigenstates (classical phase space).

In general, the choice of operators or functions for the variational ansatz is very large or even infinite, as in the classical limit. A very important insight allowing one to choose a convenient and asymptotically exact variational manifold comes from the Taylor expansion of $\partial_\lambda H(t)$,

$$\partial_\lambda H(t) = \mathrm{e}^{iHt/\hbar} \partial_\lambda H \mathrm{e}^{-iHt/\hbar} = \sum_n \frac{(it)^n}{n!} \mathcal{L}^n \partial_\lambda H$$

$$\mathcal{L}A = \frac{1}{\hbar}[H, A] \xrightarrow[\hbar \to 0]{} -i\{A, H\}. \tag{17}$$

Here $\mathcal{L}$ is the Liouvillian superoperator, which we defined including factors of $i$ and $\hbar$ such that it has a well-defined classical limit. From Eq. (13) we see that the AGP can be formally expanded in odd powers of $\mathcal{L}$ acting on $\partial_\lambda H$ [53]:

$$\mathcal{A}_\lambda = -i \sum_{k \geq 1} (-1)^k \alpha_k \mathcal{L}^{2k-1} \partial_\lambda H \quad \Rightarrow \quad G_\lambda = \sum_{k \geq 0} (-1)^k \alpha_k \mathcal{L}^{2k} \partial_\lambda H, \quad \alpha_0 = 1. \tag{18}$$

One can combine this expansion with the minimization principle (15) and treat the coefficients $\alpha_k$ as variational parameters [53–55]. It is intuitively anticipated that in integrable systems eigenstates are highly structured and should therefore be easy to deform. In classical integrable systems, trajectories are superimposed one-dimensional curves along the tori defined by action-angle variables [56]. It is expected that these tori change smoothly under integrable deformations of the Hamiltonian, and the AGP should be well defined in that context. Conversely, in chaotic systems quantum eigenstates and classical trajectories are very unstructured and unstable and should be highly susceptible to infinitesimal deformations of the Hamiltonian. Let us now quantify this intuition following Refs. [33, 35] and use the complexity of adiabatic transformations as a measure of chaos.

## 2.2 Fidelity Susceptibility as a Measure of Chaos and Ergodicity

Consider a quantum system with Hamiltonian $H$ and an eigenstate $|n\rangle \equiv |\psi_n\rangle$. Then the fidelity susceptibility, $\chi_\lambda$, is defined as the connected part of the overlap of the derivatives of this state with respect to $\lambda$ [57]:

$$\chi_\lambda^{(n)} = \langle \partial_\lambda \psi_n | \partial_\lambda \psi_n \rangle - \langle \partial_\lambda \psi_n | \psi_n \rangle \langle \psi_n | \partial_\lambda \psi_n \rangle. \tag{19}$$

Since the AGP is defined as the derivative operator, up to a factor $\hbar^2$, the fidelity susceptibility is simply the covariance of the AGP:

$$\hbar^2 \chi_\lambda^{(n)} \to \chi_\lambda^{(n)} = \langle n | \mathcal{A}_\lambda^2 | n \rangle - \langle n | \mathcal{A}_\lambda | n \rangle^2 \equiv \langle n | \mathcal{A}_\lambda^2 | n \rangle_c \tag{20}$$

As in Eq. (15) it is convenient to average $\chi_\lambda^{(n)}$ over different eigenstates with some weight $\rho_n$ and define

$$\chi_\lambda = \|\mathcal{A}_\lambda\|^2 = \sum_n \rho_n \chi_\lambda^{(n)} \equiv \sum_n \rho_n \langle n | \mathcal{A}_\lambda^2 | n \rangle_c$$

$$\xrightarrow[\hbar \to 0]{} \int d\boldsymbol{x} d\boldsymbol{p} \, P(\boldsymbol{x}, \boldsymbol{p}) \left( \mathcal{A}_\lambda^2(\boldsymbol{x}, \boldsymbol{p}) - \overline{\mathcal{A}_\lambda(\boldsymbol{x}, \boldsymbol{p})}^2 \right), \tag{21}$$

where $\overline{\mathcal{A}_\lambda(\boldsymbol{x}, \boldsymbol{p})}$ is the time average of $\mathcal{A}_\lambda(\boldsymbol{x}(t), \boldsymbol{p}(t))$ with $\boldsymbol{x}(0) = \boldsymbol{x}$, $\boldsymbol{p}(0) = \boldsymbol{p}$. Note that because the connected part of $\mathcal{A}_\lambda$ is evaluated in each eigenstate, the fidelity susceptibility is not generally a statistical variance of the operator $\mathcal{A}_\lambda$. Instead, up to a factor of four, $\chi_\lambda$ is the state-averaged quantum Fisher information [58]. In many cases (for example, when the Hamiltonian has time-reversal symmetry) one can set the Berry connections of all eigenstates to zero, $\langle n|\mathcal{A}_\lambda|n \rangle = 0$, and $\chi_\lambda$ is equivalent to the usual ensemble variance of $\mathcal{A}_\lambda$.

It follows from the discussions of Sec. 2.1 that, in chaotic systems, $\chi_\lambda$ generally diverges in the classical [45] and thermodynamic [57] limits. It also is not a self-averaging quantity for orthogonal random matrix ensembles [46] and non-random systems with time-reversal symmetry, which satisfy ETH. One approach to deal with this divergence is to define the typical fidelity susceptibility by averaging $\log \chi_\lambda$ over Hamiltonian eigenstates and exponentiating the result [34,35]. It is, however, more convenient for our purposes to use a different strategy by working with a finite frequency cutoff $\mu$ as introduced in Eqs. (12) and (13). Then the resulting $\chi_\lambda$ becomes well defined in all situations. Moreover, by analyzing the dependence of $\chi_\lambda$ on $\mu$ we can relate the fidelity susceptibility to the asymptotic long-time behavior of the operator $\partial_\lambda H$. Indeed, using Eq. (13) we find that

$$\chi_\lambda = \sum_{m,n} \rho_n \frac{\omega_{nm}^2}{\left(\omega_{nm}^2 + \mu^2\right)^2} |\langle n|\partial_\lambda H|m \rangle|^2. \tag{22}$$

This expression can be represented through an integral over the so-called spectral function $\Phi_\lambda(\omega)$:

$$\chi_\lambda = \int_{-\infty}^{\infty} d\omega \, \frac{\omega^2}{(\omega^2 + \mu^2)^2} \Phi_\lambda(\omega)$$
$$\Phi_\lambda(\omega) = \sum_n \rho_n \frac{1}{4\pi} \int_{-\infty}^{\infty} dt \, \mathrm{e}^{i\omega t} \langle n|\partial_\lambda H(t)\partial_\lambda H(0) + \partial_\lambda H(0)\partial_\lambda H(t)|n \rangle. \tag{23}$$

Note that for any $\mu > 0$ we can drop the connected part as there is no contribution to $\chi_\lambda$ from the term $m = n$. The same is true in the classical limit. The equivalence of Eqs. (22) and (23) can be established using the Lehmann representation of the spectral function. Classically, we see that $\chi_\lambda$ can be interpreted as a measure of the complexity of the canonical transformation which preserves trajectories averaged over time $1/\mu$.

From Eqs. (22) and (23) we see that the behavior of $\chi_\lambda$ is tied to the low-frequency asymptotics of the spectral function. Refs. [33,35] analyzed various one-dimensional quantum spin chains and found that there are three different generic regimes:

$$\begin{array}{lll} \textit{Ergodic/ETH}, & \Phi_\lambda(\omega \to 0) \to \text{const} > 0, & \chi_\lambda \sim \exp[S] \\ \textit{Integrable}, & \Phi_\lambda(\omega \to 0) \to 0, & \chi_\lambda \sim L^\alpha, \, \alpha \simeq 1 \\ \textit{Intermediate}, & \Phi_\lambda(\omega \to 0) \sim \omega^{-1+1/z}, \, z > 1, & \chi_\lambda \sim \exp[(2 - 1/z)S]. \end{array} \tag{24}$$

We see that the problem of small denominators (22) in quantum mechanics (equivalently the low-frequency response (23)) is directly connected to the sensitivity of quantum eigenstates; classically, it is connected to the complexity of canonical transformations which preserve time-averaged distributions. The goal of the rest of this paper is to see whether these concepts can be used to identify and characterize chaos in systems with few degrees of freedom, such as coupled rotators, which have been analyzed extensively in the literature using other probes [3,59]. In this, way we can find a unifying language suitable to distinguish chaotic, integrable and ergodic regimes both in quantum and classical systems, whether they are extended or not.

## 2.3 High Frequency Response and Operator Growth

According to conventional wisdom, classical chaos is recognizable through the complexity of trajectories. Typical particle trajectories in an integrable classical system

are highly strctured, while typical chaotic trajectories appear random. It is therefore unnecessary to compare two different trajectories to visually recognize chaos. One approach to studying classical trajectories, with natural extensions to quantum systems, is the short-time expansion. As an example, consider a classical nonlinear oscillator in two dimensions with Hamiltonian given by Eq. (3) where we additionally set $\omega_1 = \omega_2 = 0$

$$H = \frac{p_1^2}{2} + \frac{p_2^2}{2} + \frac{(x^2 + y^2)}{2} + \frac{\epsilon}{4} x^2 y^2, \tag{25}$$

which is generally chaotic for $\epsilon > 0$. The corresponding equations of motion are

$$\frac{dx_1}{dt} = p_1, \quad \frac{dp_1}{dt} = -x_1 - \frac{\epsilon}{2} x_1 x_2^2, \tag{26}$$

$$\frac{dx_2}{dt} = p_2, \quad \frac{dp_2}{dt} = -y x_2 - \frac{\epsilon}{2} x_1^2 x_2. \tag{27}$$

This is a system of nonlinear partial differential equations, so a general analytic solution does not exist. Nevertheless, we can carry out a short-time expansion of the solution to arbitrary order in $t$:

$$x_1(t) = x_1 + p_1 t - \frac{2x_1 + \epsilon x_1 x_2^2}{4} t^2 - \frac{2p_1 + \epsilon(2p_2 x_1 x_2 + p_1 x_2^2)}{24} t^3$$
$$+ \frac{4x_1 + \epsilon(8x_1 x_2^2 - 4x_1 p_2^2 - 8p_1 p_2 x_2) + \epsilon^2(2x_1^3 x_2^2 + x_1 x_2^4)}{1152} t^4 + \dots \tag{28}$$

We can view this expansion as a map from an initially simple function $x_1$ to progressively more complex functions describing a trajectory. Clearly, the complexity of these functions increases with each order of the expansion when $\epsilon \neq 0$.

Now let us perform a similar analysis for the corresponding quantum problem by expanding the operator $\hat{x}_!(t)$ in the Heisenberg picture; we use hats to distinguish operators from the corresponding classical variables.

$$\hat{x}_1(t) \equiv e^{\frac{i}{\hbar}\hat{H}t} \hat{x}_1 e^{-\frac{i}{\hbar}\hat{H}t} = \hat{x}_1 + \hat{p}_1 t - \frac{2\hat{x}_1 + \epsilon \hat{x}_1 \hat{x}_2^2}{4} t^2 - \frac{2\hat{p}_1 + \epsilon(\hat{x}_1\{\hat{p}_2, \hat{x}_2\}_+ + \hat{p}_1 \hat{x}_2^2)}{24} t^3$$
$$+ \frac{4\hat{x}_1 + \epsilon(8\hat{x}_1 \hat{x}_2^2 - 4\hat{x}_1 \hat{p}_2^2 - 4\hat{p}_1\{\hat{p}_2 \hat{x}_2\}_+) + \epsilon^2(2\hat{x}_1^3 \hat{x}_2^2 + \hat{x}_1 \hat{x}_2^4)}{1152} t^4 + \dots \tag{29}$$

where $\{\hat{A}, \hat{B}\}_+ = \hat{A}\hat{B} + \hat{B}\hat{A}$. We see that the expansions (28) and (29) are quite similar [60, 61]. In fact, if we use the Weyl correspondence to map quantum operators to functions of phase space variables [60, 61] the two expansions are identical at this order. That is, the Weyl symbol of $\hat{x}_1(t)$, denoted $(\hat{x}_1(t))_{\text{W}}$,

$$(\hat{x}_1(t))_{\text{W}} = \int \int d\boldsymbol{\xi} \left\langle \mathbf{x} - \frac{\boldsymbol{\xi}}{2} \middle| \hat{x}_1(t) \middle| \mathbf{x} + \frac{\boldsymbol{\xi}}{2} \right\rangle e^{\frac{i}{\hbar}\mathbf{p}\boldsymbol{\xi}},$$

with $\mathbf{x} = (x_1, x_2)$, $\mathbf{p} = (p_1, p_1)$, $\boldsymbol{\xi} = (\xi_1, \xi_2)$, reproduces the expansion (28). Hence, while trajectories in quantum systems are ill-defined, there is a well-defined procedure for defining the short time expansion of a Heisenberg operator, which can be associated with a quantum trajectory. In fact, the exact correspondence between quantum and classical short-time expansions holds to order $t^6$. For the Hamiltonian (25) the difference of these expansions at sixth order is given by [3]

$$(\hat{x}_1(t))_W - x_1(t) = -\frac{\epsilon^2 \hbar^2 t^6}{2880} x_1 + O(t^7). \tag{30}$$

---

[3]For a general potential $V(x_1, x_2)$, one can show that $(\hat{x}_1(t))_W - x_1(t)$ is given by

$$-\frac{\hbar^2 t^6}{2880} \left( \frac{\partial^3 V}{\partial x_2^3} \frac{\partial^4 V}{\partial x_1 \partial x_2^3} + 3\frac{\partial^3 V}{\partial x_1 \partial x_2^2} \frac{\partial^4 V}{\partial x_1^2 \partial x_2^2} + 3\frac{\partial^3 V}{\partial x_1^2 \partial x_2} \frac{\partial^4 V}{\partial x_1^3 \partial x_2} + \frac{\partial^3 V}{\partial x_1^3} \frac{\partial^4 V}{\partial x_1^4} \right) + O(t^7)$$

We see that analyzing short time dynamics allows us to effectively compare quantum and classical dynamics directly, irrespective of initial conditions. Of course, one also needs to keep in mind that the dynamics of physical observables does depend on the initial state of the system, which can be very different for quantum and classical systems, for example, at low temperatures.

Let us now analyze the short-time expansion more systematically. Instead of looking into a particular phase space variable $x(t)$, we can analyze an arbitrary observable $\mathcal{O}(\boldsymbol{x}, \boldsymbol{p})$. To connect with Secs. 2.1 and 2.2, we write $\mathcal{O} \equiv \partial_\lambda H$ such that $\lambda$ is a source for the observable $\mathcal{O}$. In the Heisenberg picture, the short-time expansion is given by Eq. (17), and the objects which appear in the short time expansion are proportional to nested commutators of the observable $\partial_\lambda H$ and the Hamiltonian $H$. In classical systems, using the fact that

$$\frac{1}{\hbar}[A, B] \to i\{A, B\}$$

we obtain a similar expansion with nested Poisson brackets in place of nested commutators. We can therefore think of short-time expansions in terms of classical function spreading or quantum operator growth. The intuition behind these names is clear from our toy example (29), where we saw that with each order of the expansion the initial "simple" operator $\hat{x}$ spreads in the space of functions spanned by higher and higher order polynomials of phase space variables. In interacting many-particle systems, this growth with each order of the expansion also involves an increasing number of degrees of freedom. Intuitively, one can expect that in chaotic systems operators spread faster than in integrable systems, as in the latter there are typically constraints imposed by various selection rules and additional conservation laws.

This intuition was formalized in several recent works [13–16] following the pioneering work of Parker et. al. [12]. They argued that for generic spin systems with local interactions, operators which are not conserved grow as rapidly as possible, consistent with fundamental constraints such as spatial locality. In particular, it was conjectured that at large $k$

$$R_k^2 \equiv \sum_{n, m \neq n} \rho_n |\langle n | \mathcal{L}^k \partial_\lambda H | m \rangle|^2 \sim \frac{(2k)!}{\tau^{2k+1}}, \tag{31}$$

where $\tau$ is a model-dependent parameter. In one dimensional systems this growth is reduced by a logarithmic correction for geometric reasons [14, 15]. This asymptotic form is expected to hold for any ensemble $\{\rho_n\}$. At the same time, it was conjectured and numerically confirmed that operator growth is reduced in integrable spin chains.

As discussed in Sec. 2.1, nested commutators naturally appear when studying adiabatic transformations and in the variational construction of the AGP. These objects are also closely related to the spectral function; using the fact that

$$\langle n | \mathcal{L}^k \partial_\lambda H | m \rangle = \omega_{nm}^k \langle n | \partial_\lambda H | n \rangle$$

we find

$$R_k^2 = \int d\omega \, \omega^{2k} \Phi_\lambda(\omega) \tag{32}$$

Comparing this expression with Eq. (23), we see that $\chi_\lambda$ is nothing but a regularized form of $R_{-1}^2$. It is also straightforward to see that the factorial growth of the norms $R_k^2$ is equivalent to exponential decay of the spectral function at large frequencies [12, 13]:

$$\Phi_\lambda(\omega) \sim \mathrm{e}^{-\tau \omega}, \quad \omega \to \infty \tag{33}$$

Indeed with this asymptote Eq. (32) reduces to the $\Gamma$-function with the correct factorial scaling. It was confirmed numerically that in integrable spin chains the spectral function $\Phi_\lambda(\omega)$ decays faster than exponentially at high frequencies, corresponding to slower operator growth. For example, in free models such as the transverse field Ising model [62], $\Phi_\lambda(\omega) = 0$ for $\omega > \omega_J$, which is the largest single-particle energy scale. This scaling corresponds to the exponential growth of nested commutators,

$R_k^2 \sim \omega_J^{2k+1}$. For some interacting integrable models the decay of $\Phi_\lambda(\omega)$ is more consistent with Gaussian decay at large $\omega$ [17,18]. This Gaussian decay also corresponds to a slower operator growth $R_k^2 \sim \sqrt{(2k)!}/\tau^{2k}$.

These results are modified in classical models, where it was argued in Ref. [19] that even in integrable systems the presence of a saddle leads to factorial growth of $R_k$. In this work, we will argue that even without any saddles such factorial behavior of $R_k$ is generic irrespective of whether the system is integrable or chaotic and thus cannot serve as an indicator of chaos. Therefore the issue of connecting trajectories defined through short time expansions to chaos remains an open problem, both in quantum and classical systems.

## 3 The General Model

In the remainder of this paper, we analyze a model of two interacting spin-$S$ degrees of freedom, $\boldsymbol{S}_1$ and $\boldsymbol{S}_2$. For finite $S$ these are described by the spin operators $\hat{\boldsymbol{S}}$ which satisfy the commutation relations $\left[\hat{S}_l^\alpha, \hat{S}_j^\beta\right] = i\hbar\delta_{lj}\epsilon_{\alpha\beta\gamma}\hat{S}_j^\gamma$, where $\alpha = \{x,y,z\}$, $l,j = \{1,2\}$, and $\epsilon_{\alpha\beta\gamma}$ is the Levi-Civita symbol. In terms of these operators, our model Hamiltonian is given by

$$\hat{H} = \hbar^2 \sum_\alpha \left[ -J_\alpha \hat{S}_1^\alpha \hat{S}_2^\alpha + \frac{1}{2}A_\alpha \left( (\hat{S}_1^\alpha)^2 + (\hat{S}_2^\alpha)^2 \right) \right] \tag{34}$$

with arbitrary couplings $\boldsymbol{J}, \boldsymbol{A}$. It is convenient to set $\hbar = 1/\sqrt{S(S+1)}$ so that in the classical limit $S \to \infty$ the equations of motion are well-defined [59]. In the classical limit, the canonical commutation relations are replaced by Poisson brackets of the

Integrable                  Chaotic

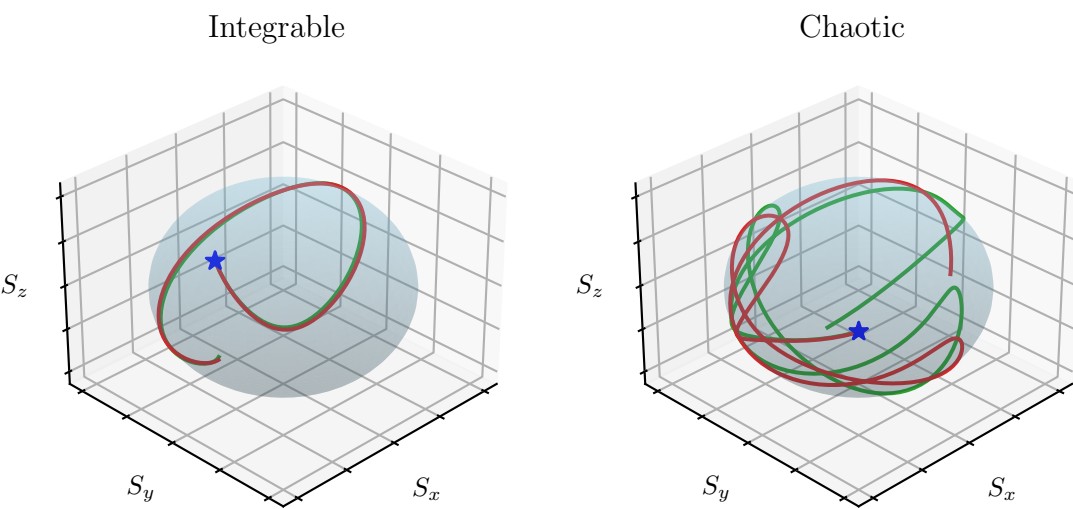

Figure 3: Visualization of the classical spin $\boldsymbol{S}_1(t) \equiv \boldsymbol{S}(t)$ for two similar initial conditions in the integrable XXZ model (left) with couplings $\boldsymbol{J} = (1,1,1/2)$, $\boldsymbol{A} = 0$ and a chaotic model with couplings $\boldsymbol{J} = (3/2, \pi, \sqrt{e})$, $\boldsymbol{A} = (\sqrt{\pi}, \sqrt{3}, e)$ (right). At $t = 0$, the two initial conditions depart from the same position (blue stars) with slightly different momenta. In the integrable case the trajectories are highly structured and remain close, while the chaotic trajectories rapidly separate over the same time interval, exhibiting a strong sensitivity to initial conditions.

corresponding spin variables $\boldsymbol{S}_j$ lying on a unit sphere:

$$\{S_l^\alpha, S_j^\beta\} = \delta_{lj}\epsilon_{\alpha\beta\gamma}S_j^\gamma, \quad \sum_\alpha (S_j^\alpha)^2 = 1 \tag{35}$$

The corresponding classical Hamiltonian is then

$$H = \sum_\alpha \left[ -J_\alpha S_1^\alpha S_2^\alpha + \frac{1}{2}A_\alpha \left( (S_1^\alpha)^2 + \left(S_2^\alpha\right)^2 \right) \right], \tag{36}$$

with the equations of motion

$$\frac{d\mathbf{S}_l}{dt} = \{\mathbf{S}_l, H\} = -\mathbf{S}_l \times \frac{\partial H}{\partial \mathbf{S}_l}. \tag{37}$$

This model has been studied previously both in the quantum and classical limits [3, 59] and is known to be integrable when the couplings satisfy

$$(A_x - A_y)(A_y - A_z)(A_z - A_z) + \sum_{\alpha\beta\gamma=cycl(xyz)} J_\alpha^2(A_\beta - A_\gamma) = 0 \tag{38}$$

and chaotic otherwise. In particular, the model is always integrable in the absence of single-ion anisotropy, $A_x = A_y = A_z = \text{const}$. In Fig. 3 we show characteristic classical trajectories for the components of $\boldsymbol{S}_1$ in both chaotic and integrable regimes which demonstrate the extreme sensitivity of chaotic trajectories to small changes of initial conditions.

In the remainder of this section we discuss details of computing the spectral function and fidelity susceptibility, which are the key objects in our analysis both for finite S and in the classical limit $S \to \infty$.

## 3.1 Methods of Analysis

Our primary goal throughout this work is to demonstrate that spectral functions and the related fidelity susceptibility can be used to detect different dynamical regimes. For Hamiltonians of the form (36), a nice choice of observable is given by the product of spin $z$-components, $\partial_\lambda H = \hbar^2 \hat{S}_1^z \hat{S}_2^z \equiv ZZ$. Importantly, this observable is integrability-preserving in the sense that if $\boldsymbol{A} = 0$, the deformed Hamiltonian $H(\lambda+\delta\lambda)$ remains integrable, i.e., it satisfies the condition (38). Such integrable observables yield the most sensitive probes of chaos as they rapidly change their long time response when integrability conditions are slightly broken [33, 41].

In the following, we compute and analyze the autocorrelation function $C_{ZZ}(t)$, associated spectral function $\Phi_{ZZ}(\omega)$ and fidelity susceptibility $\chi(\mu)$,

$$\begin{aligned} C_{ZZ}(t) &= \frac{1}{\mathcal{D}}\text{Tr}\left[ZZ(t)ZZ(0)\right] \xrightarrow[\hbar\to 0]{} \overline{ZZ(t)ZZ(0)} \\ \Phi_{ZZ}(\omega) &= \frac{1}{2\pi}\int_{-\infty}^{\infty} dt\, e^{i\omega t}C_{ZZ}(t) \\ \chi(\mu) &= \int_{-\infty}^{\infty} d\omega\, \frac{\omega^2}{\left(\omega^2 + \mu^2\right)^2}\Phi_{ZZ}(\omega) \end{aligned} \tag{39}$$

where $\mathcal{D}$ is the Hilbert space dimension and overlines indicate uniform phase space averages over classical initial conditions.

In the following, we will be interested in extracting both the high and low frequency asymptotics of the spectral function. As discussed in Sec. 2, the former is related to operator spreading, and the latter to the fidelity susceptibility. For each of the models considered in this paper, we find that $\Phi_{ZZ}(\omega \to \infty) \sim e^{-\tau\omega}$, which makes the analysis of that regime particularly straightforward. The decay rate of the spectral function can be computed directly from nested commutators or Poisson brackets (see Eq. (31) and surrounding discussion).

The low frequency behavior of $\Phi_{ZZ}(\omega)$ is both richer and more subtle. For the integrable systems that we have studied (Sections 4, 5), the spectral function vanishes in the limit $\omega \to 0$, leading to a finite fidelity susceptibility $\chi$ in the limit $\mu \to 0$ for all values of $S$ (see Figs. 5, 6). In the chaotic regime (Sec 6), the spectral function decays weakly as $\omega \to 0$ at intermediate values of $S$, while in the classical limit $S \to \infty$ it diverges instead as $\omega \to 0$, leading to a fidelity which grows rapidly as $\mu \to 0$ (see Fig. 8). In the chaotic mixed phase-space regime, the low-frequency behavior of the spectral function and the dependence of $\chi(\mu)$ are very different in regular and chaotic regions if, instead of the uniform phase space averaging in (39), we perform averaging over specific trajectories.

With the exception of the XXZ model (Sec. 4), where a closed-form solution is available in the classical limit, direct analysis of $\Phi_{ZZ}(\omega)$ requires the use of numerical simulations. We carry out these computations using exact diagonalization for finite $S$ and nonlinear dynamics simulations of the equations of motion in the classical limit. In the next two sections, we review the computational techniques employed in both the quantum and classical analyses in sufficient detail to reproduce our results.

### 3.1.1 Quantum Methods

In quantum systems, the computation of $\Phi_{ZZ}(\omega)$ is accomplished most simply via exact diagonalization. To this end, it is useful to write the spectral function directly in the Lehmann representation,

$$\Phi_{ZZ}(\omega) = \frac{1}{\mathcal{D}} \sum_{n,m} |\langle n|ZZ|m\rangle|^2 \delta(\omega - \omega_{nm}). \tag{40}$$

The reader is reminded that factors of $\hbar$ are important to keep in mind when taking the classical limit. The computation of $\Phi_{ZZ}(\omega)$ is then simple in principle: one just needs the eigenvalues and eigenvectors of the Hamiltonian.

The spectral function which results from this procedure is a very dense and noisy collection of delta functions. To extract smooth features of the spectral function, we perform a filtering procedure which replaces delta functions with Gaussians whose widths are set by the typical level spacing of the Hamiltonian, denoted by $\delta$. A smoothened estimator of the spectral function then follows by summing over these Gaussian-weighted contributions,

$$\Phi_{ZZ}(\omega) \approx \frac{1}{\sqrt{\pi}\delta\mathcal{D}} \sum_{n,m} |\langle n|ZZ|m\rangle|^2 \exp\left[\frac{(\omega - \omega_{nm})^2}{\delta^2}\right] \tag{41}$$

In addition to this filtering procedure, we have found that it is useful to introduce weak (on the order of 3%) disorder into the Hamiltonian couplings and average the spectral function over disorder. This minimizes the role of accidental resonances which may be present in a particular Hamiltonian and generally leads to smoother results; however, care must be taken to guarantee that the disorder does not break integrability conditions or symmetries of the model, if present. We will return to this point in subsequent model-specific sections.

### 3.1.2 Classical Methods

To compute the spectral function in the classical limit, we directly simulate trajectories using the equations of motion (37). The first step in this process is to generate a set of initial conditions. For this purpose we make use of Sobol sequences to construct a quasi-uniform distribution over the classical phase space, which is known to lead to faster convergence than a naive uniform distribution [63]. In the case of our two-spin model, we generate four length $N$ Sobol sequences which represent the initial spin angles $\{\theta_i, \phi_i\}$. In addition, we follow a well-established heuristic in the literature [64, 65] that the first $\log_2(N/2) - 1$ entries of each Sobol sequence are discarded. In

practice, we find that this eliminates highly symmetric initial conditions, which can introduce numerical artifacts by producing, for example, stationary configurations. We also note in passing that the sampled observables must be multiplied by the product $\sin\theta_1 \sin\theta_2$ coming from the integration measure.

Once these initial conditions are generated, we solve equations of motion both forward and backward in time to find $\boldsymbol{S}_i(t)$ and $\boldsymbol{S}_i(-t)$. This allows us to restrict integration of the autocorrelation function in the time domain of Eq. (39) to $t \geq 0$. To solve the equations of motion, we employ Runge-Kutta methods [66] to compute the spin trajectories $\boldsymbol{S}_1(t), \boldsymbol{S}_2(t)$ out to time $T$ in time steps of size $dt$. From each trajectory, we extract its contribution to the autocorrelation function, $ZZ(t)ZZ(0)$, and the Fourier transform

$$\frac{1}{2\pi} \int_{-T}^{T} dt \; ZZ(t)ZZ(0)e^{i\omega t - \alpha^2 t^2} \tag{42}$$

where $\alpha^{-1}$ is a late-time cutoff which smoothly controls the error introduced by the fact that $T$ is finite. Unless otherwise noted, we fix $\alpha = 5/T$ for the remainder of the paper and always check that our results are $\alpha$-independent.

Formally, averaging Eq. (42) over initial conditions yields the spectral function $\Phi_{ZZ}(\omega)$ for frequencies which are sufficiently large compared to $\alpha$. In practice, we have found that this average converges extremely slowly since, while the spectral function is strictly positive, the contribution of a particular initial condition need not be. This slow convergence can be remedied by using the fact that, after averaging over initial conditions, the autocorrelation function is invariant under time translations due to the spectral theorem:

$$\overline{ZZ(t)ZZ(0)} = \overline{ZZ(t+\tau)ZZ(\tau)}, \quad \forall \tau \tag{43}$$

Hence the spectral function is given by

$$\Phi_{ZZ}(\omega) = \frac{1}{2\pi} \overline{|ZZ(\omega)|^2}, \quad ZZ(\omega) = \frac{1}{2\pi} \int_{-\infty}^{\infty} dt \; ZZ(t)e^{i\omega t - \alpha^2 t^2}. \tag{44}$$

Expressed in this way, each trajectory's contribution to the spectral function is manifestly positive and our numerical computations have found that this average has significantly improved convergence properties.

# 4 The XXZ Model

In this section, we analyze the two-spin XXZ model described by the Hamiltonian (36) with couplings $A_\alpha = 0$ and $(J_x, J_y, J_z) = (J_\perp, J_\perp, J_z)$. In the classical limit, this model has a closed-form solution for the observable $ZZ(t)$ which allows us to extract the high and low frequency behavior of $\Phi_{ZZ}(\omega)$; these results are presented in full detail in App. A. Here, we outline a simple intuitive picture which captures the qualitative behavior of the spectral function at high and low frequencies. In addition, we present numerical comparisons of the quantum and classical spectral functions at all frequencies.

The XXZ model has a U(1) symmetry associated with rotations about the $z$-axis; correspondingly, the magnetization $M_z = (S_1^z + S_2^z)/2$ is conserved. In combination with the energy $E$ there are then two conserved quantities, guaranteeing that the two-spin XXZ model is integrable for all spin magnitudes $S$ including the classical model. In the classical limit, the observable $ZZ(t)$ can be expressed in terms of $M_z$ by introducing another coordinate $z = (S_1^z - S_2^z)/2$,

$$ZZ(t) = \frac{1}{2}\left(M_z^2 - z(t)^2\right). \tag{45}$$

The problem of analyzing $ZZ(t)$ is therefore reduced to that of $z(t)$. Remarkably, $z(t)$ can be interpreted as the coordinate of a particle with unit mass in a

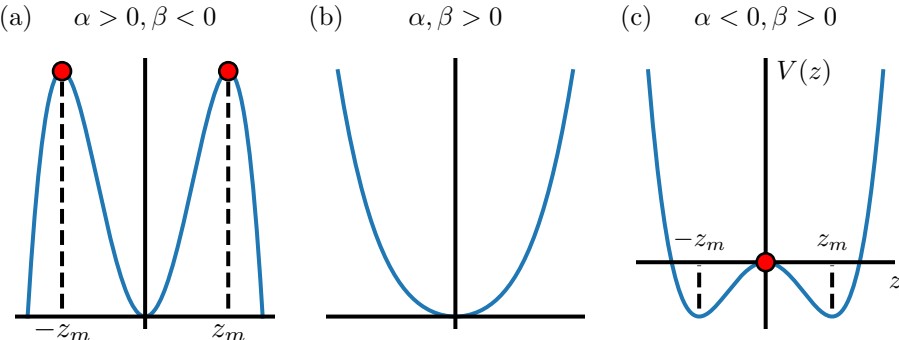

Figure 4: The effective one-dimensional potential, $V(z)$, as a function of the co-ordinate $z = (S_1^z - S_2^z)/2$. The parameters $\alpha$ and $\beta$ which enter $V(z)$ are defined in Eq. (46). The period of a trajectory diverges as its turning point approaches a local maximum of $V(z)$ (red circles), which controls the low frequency behavior of $\Phi_{ZZ}(\omega)$. The maxima are accessible in the easy axis regime (a) and easy plane regime at low energies (c), while no maxima are present in the easy plane regime at high energies (b). At the Heisenberg point (not shown) the quartic term vanishes and the low frequency regime is controlled by trajectories with small $\alpha$.

one-dimensional quartic potential, a result derived in Ref. [59]. More precisely, $z(t)$ satisfies the differential equation $\ddot{z}(t) = -V'(z)$, where $V(z) = \alpha z^2 + \beta z^4$ and

$$\alpha = J_\perp^2 - J_z E + \left(J_\perp^2 - J_z^2\right) M_z^2$$
$$\beta = \frac{1}{2}\left(J_z^2 - J_\perp^2\right) \tag{46}$$
$$E_{\text{eff}} = \frac{J_\perp^2}{2}(1 - M_z^2)^2 - \frac{1}{2}(E + J_z M_z^2)^2$$

where $E_{\text{eff}}$ is the effective energy of the 1D particle. Notice that $\beta$ is a function of the Hamiltonian couplings alone: it is positive in the easy axis regime, negative in the easy plane regime and zero at the Heisenberg point, while $\alpha$ and $E_{\text{eff}}$ depend on initial conditions through the integrals of motion.

Schematic plots of the potential $V(z)$ are shown in Fig. 4 for several choices of parameters, demonstrating the possible sign combinations of $\alpha$ and $\beta$. These potentials offer a simple heuristic to estimate both the high and low frequency behavior of $\Phi_{ZZ}(\omega)$. We begin by expressing the spectral function explicitly in terms of the trajectories $z(t)$,

$$\Phi_{ZZ}(\omega) = \overline{\frac{1}{2\pi}\int dt\, e^{i\omega t}\, C_{ZZ}(t)} = \overline{ZZ(0)\left(M_z^2\,\delta(\omega) - \frac{1}{2\pi}\int dt\, e^{i\omega t}\, z(t)^2\right)}. \tag{47}$$

Each trajectory is a periodic function of time which is either symmetric about $z = 0$ ($E_{\text{eff}} > 0$) or localized away from $z = 0$ ($E_{\text{eff}} < 0$). For the former, the fundamental frequency of $z(t)^2$ is twice that of $z(t)$ while for the latter they are identical. In either case, low-frequency contributions to $\Phi_{ZZ}(\omega)$ come from trajectories with long periods.

The set of long-period trajectories is naturally broken into two categories. The first are those with small values of the couplings $\alpha, \beta \ll 1$ and a small amplitude of oscillation, which are approximately harmonic. In general, these trajectories are irrelevant to the asymptotes of $\Phi_{ZZ}(\omega)$ because there are usually other trajectories which contribute at more extreme frequencies due to anharmonic effects. The exception is at the Heisenberg point, where harmonic trajectories dominate due to the absence of a quartic term (see App. A.2.1).

The second category of long-period trajectories are those with turning points near the local maxima of $V(z)$, denoted $z_m$, which appear when $\alpha$ and $\beta$ have opposite

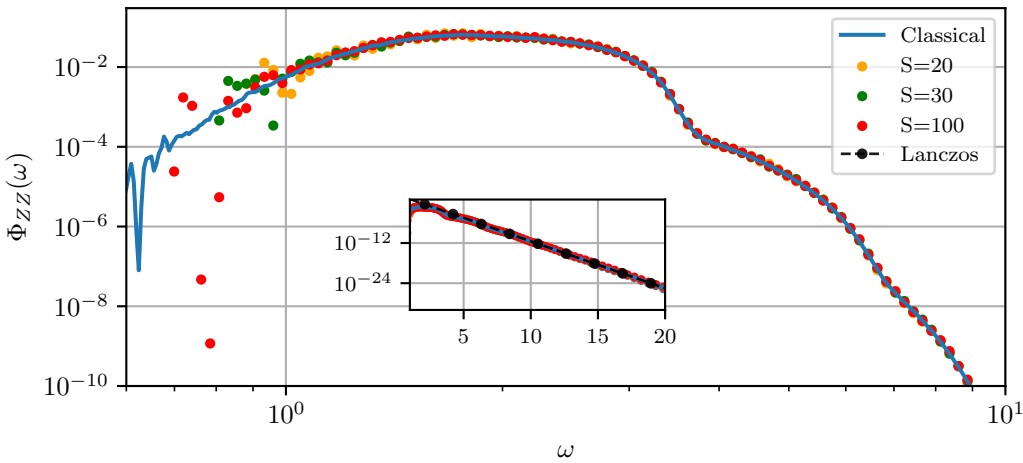

Figure 5: Spectral functions of the XXZ model with parameters $\boldsymbol{J} = (1, 1, 1/2)$. The classical data was obtained by averaging $N = 3 \times 10^6$ analytically exact trajectories (see App. A) with Sobol-random initial conditions. The quantum data was obtained by averaging over $N = 10^3$ disorder realizations with Hamiltonian couplings $\boldsymbol{J} + \delta J (1, 1, 1/2), |\delta J| \leq 0.03$. Inset: the high frequency regime decays exponentially with a rate $\tau \approx 3.23$ consistent with computations from the Lanczos approach discussed in App. B.

signs (Fig. 4a, c). Since these maxima are unstable equilibria, the period is expected to be exponentially sensitive to deviations from the maxima, such that $T \sim \ln\left(|z_m - z(0)|^{-1}\right)$. Hence low frequency contributions to $\Phi_{ZZ}(\omega)$ require exponentially precise fine-tuning of initial conditions and it is natural to conjecture that $\Phi_{ZZ}(\omega \to 0) \sim \exp\left[-1/\text{poly}(\omega)\right]$. The results of App. A.2 confirm this intuition; more precisely, we argue that

$$\Phi_{ZZ}(\omega \to 0) \lesssim \begin{cases} \exp\left[-f(J_\perp, J_z)/\omega\right] & J_\perp > J_z \\ \omega/(64\pi J^2) & J_\perp = J_z \\ \exp\left[-g(J_\perp, J_z)/\sqrt{\omega}\right] & J_\perp < J_z \end{cases} \tag{48}$$

for some functions $f, g$ of the couplings alone. In each case, the low-frequency weight of $\Phi_{ZZ}$ drops off as $\omega \to 0$, consistent with our expectations for integrable systems. The spectral weight is greatest at the Heisenberg point, where it vanishes only linearly with $\omega$. Even in this case, the fidelity diverges only logarithmically, $\chi(\mu) \sim \log(\mu)$. Note that this weak divergence of $\chi$ is consistent with our general expectations as the Hamiltonian deformation $H \to H + ZZ \delta\lambda$ breaks an $SU(2)$ symmetry, lifting associated degeneracies in the spectrum and hence increasing the low-frequency response.

The high frequency behavior of $\Phi_{ZZ}(\omega)$ can also be qualitatively understood using the potentials of Fig. 4. Since each trajectory $z(t)$ is a solution of Newton's equation, the spectral function is an average of analytic functions and we expect $\Phi_{ZZ}(\omega)$ decays at least exponentially quickly as $\omega \to \infty$. In App. A.3, we show that the decay of the spectral function is precisely exponential except at the Heisenberg point, where the absence of anharmonic terms leads to a sharp cutoff, $\Phi_{ZZ}(\omega > 2J) \equiv 0$. More precisely, we argue that the Fourier transform of each trajectory decays exponentially at high frequencies with a rate $\tau[z(t)]$. We evaluate the decay rate in closed form for trajectories which are dominated by either the quartic or quadratic parts of $V(z)$:

$$\tau[z(t)] \sim \begin{cases} \ln|\beta|/\sqrt{2\alpha}, & |\beta| \ll \alpha \\ (\beta E_{\text{eff}})^{-1/4}, & \alpha = 0, \beta > 0. \end{cases} \tag{49}$$

We denote the minimum of $\tau[z(t)]$ over all trajectories by $\tau_0$, which sets the decay rate of the full spectral function. At low energies, the quadratic part of $V(z)$ dominates

the potential and trajectories behave as though $|\beta| \ll \alpha$. Eq. (49) then predicts that the spectral function at high frequencies decays exponentially with a rate $\tau_0 \sim \ln|\beta|$ which diverges logarithmically as we approach the Heisenberg point (see Fig. 20). This divergence is consistent with the fact that the spectral function at the Heisenberg point has a hard high-frequency cutoff.

In Fig. 5, we compare quantum calculations of $\Phi_{ZZ}(\omega)$ with classical results computed by averaging analytically exact trajectories $z(t)$ with randomly drawn initial conditions. The agreement between both sets of data is very precise, although errors due to sampling effects become pronounced at low frequencies.

The derivation of Eq. (49) is made possible by the simplicity of the XXZ model in the classical limit. A more general approach to computing the high-frequency decay of spectral functions comes via nested commutators or Poisson brackets. As discussed in Sec. 2.3, the moments $R_n^2$ of the spectral function are given by norms of nested commutators (see Eq. (31) and surrounding discussion). Assuming that $\Phi_{ZZ}(\omega \to \infty) \sim e^{-\tau_0 \omega}$, the decay rate $\tau_0$ can be estimated as

$$\tau_0 \approx \sqrt{\frac{R_n^2(2n+2)(2n+1)}{R_{n+1}^2}} \tag{50}$$

This approximation improves with increasing $n$, as the weight of the moments is increasingly concentrated in the tail of the spectral function. In practice, this estimate converges rapidly as a function of $n$, and the inset of Fig. 5 shows that the predicted decay rate is highly accurate. Additional comments connecting nested commutators and Poisson brackets with the Lanczos formalism are presented in App. B.

## 5 The XYZ Model

In this section we consider the XYZ model, described by the Hamiltonian (36) with $\boldsymbol{A} = 0$, $J_x \neq J_y \neq J_z$. This model satisfies the integrability condition Eq. (38)

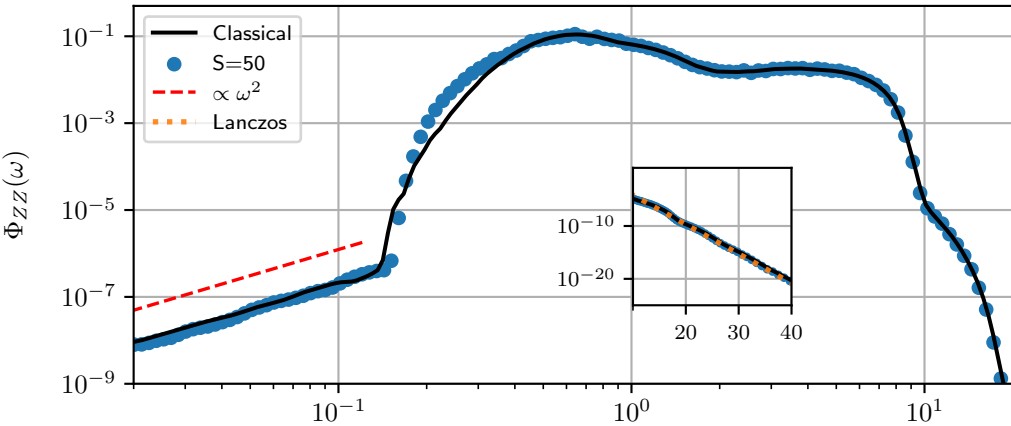

Figure 6: Spectral functions of the XYZ model. The classical data is computed by simulating trajectories to time $T = 2 \times 10^3$ in time steps of size $dt = 5 \times 10^{-3}$ with cutoff $\alpha = 7/T$ and averaging over 5 disorder realizations with $N = 10^6$ trajectories in each realization. The quantum data averages over $10^3$ disorder realizations. Fits to the low frequency regime show $\Phi(\omega \to 0) \sim \omega^2$. Inset: the high-frequency regime exhibits exponential decay of the spectral function. The decay rate $\tau \approx 1.281$ is computed with nested Poisson brackets, shown by the curve labeled Lanczos.

for any choice of $\boldsymbol{J}$ but does not have any continuous symmetries and an analytic solution appears intractable. We choose couplings which are positive incommensurate numbers $\boldsymbol{J} = (\sqrt{\pi}, 5/2, e)$, but their precise values are inessential to our qualitative results. To reduce the effects of resonances in the quantum model, we also introduce a weak disorder in the couplings, $J_\alpha \to J_\alpha(1 + r_\alpha)$, where $r_\alpha$ is drawn uniformly from the range $[-0.03, 0.03]$.

Spectral functions for the XYZ model are shown in Fig. 6. The agreement between quantum and classical results is strong, even for a relatively small $S$ on the order of 50. Numerical fits indicate that the spectral function decays as a power law at low frequencies, $\Phi_{ZZ}(\omega \to 0) \sim \omega^2$, and the fidelity susceptibility is finite in the limit $\mu \to 0$. We see that the low frequency tail goes to zero slower than in the more symmetric XXZ model but faster than for the XXX model, where the observable $ZZ$ breaks the $SU(2)$ symmetry. Interestingly, this low frequency asymptote is preceded by a sharp decay of the spectral function to a very small value, similar to the XXZ model (see Fig. 5). The high frequency behavior of $\Phi_{ZZ}(\omega)$ is also shown in Fig. 6 (inset). There the spectral function decays exponentially, with a rate which is accurately predicted by the norms of nested Poisson brackets.

# 6 Chaotic Model: Universality in the Emergence of Chaos

In this section, we shift focus away from integrable systems and study the spectral properties of a chaotic model. Classical chaotic dynamics are characterized by complex trajectories which respond very strongly to Hamiltonian deformations. Per the arguments of Sec. 2, the strong sensitivity of time-averaged trajectories to small perturbations is captured by the presence of low frequency weight in the spectral functions of generic physical observables and corresponding growth of the fidelity susceptibility as the late-time cutoff, $\mu$, decreases.

The chaotic model that we will consider is described by the Hamiltonian (36) with couplings

$$\boldsymbol{J} = (3/2, \pi, \sqrt{e}), \quad \boldsymbol{A} = x(\sqrt{\pi}, \sqrt{3}, e) \tag{51}$$

where $x$ is a tunable parameter which characterizes the strength of the integrability

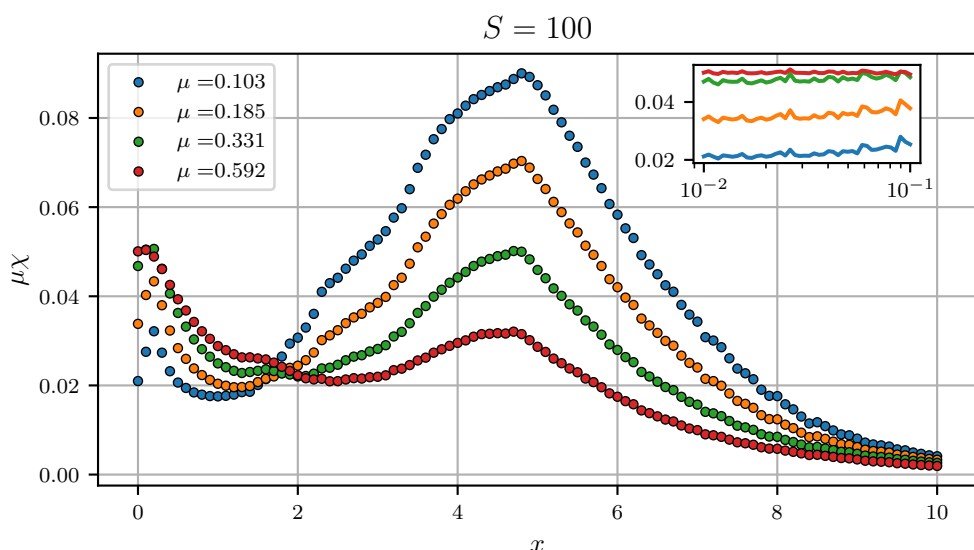

Figure 7: Rescaled fidelity, $\mu\chi$, as a function of the integrability breaking parameter $x$ with $S = 100$. Inset: the rescaled fidelity in the small $x$ regime.

breaking perturbation. This model is integrable for $x = 0$ and $x \to \infty$ (see Eq. (38)) and chaotic for any finite nonzero $x$, which makes it a useful testing ground for the fidelity susceptibility as a probe of different dynamical regimes. Moreover, as we will see, the crossover between quantum and classical dynamics is very rich. All quantum data reported in this section averages over disorder by mapping the Hamiltonian couplings $J_\gamma \to (1 + r_\gamma)J_\gamma$, where each $r_\gamma$ is a uniform random variable drawn from the interval $[-0.03, 0.03]$.

To develop an intuition for the different regimes in this model, consider Fig. 7, which reports the rescaled fidelity $\mu\chi$ as a function of $\mu$ and $x$ at fixed $S = 100$. This scaling is useful because it allows one to clearly distinguish between integrable, maximally chaotic and ergodic regimes as schematically illustrated in Fig. 1, which are in turn related to different low frequency asymptotics of the spectral function (see Eq. (24) and surrounding discussion). In particular, in the integrable regime where $\chi(\mu \to 0) \to$ const, the rescaled fidelity $\mu\chi$ vanishes as $\mu \to 0$. This is indeed what we observe at small values of $x$ (Fig. 7 inset).

On the other hand, ergodic systems have spectral functions which saturate at small frequencies such that $\mu\chi$ approaches a constant as $\mu \to 0$. Similar behavior of the spectral function is observed at integrable points if the observable breaks integrability [33, 41, 67]. The data of Fig. 7 is consistent with the latter scenario at large $x$. Outside of this limit, $\mu\chi$ does not saturate and there is no clear domain of ergodicity. Instead, for intermediate values of $x$ ($2 \lesssim x \lesssim 8$), $\chi(\mu)$ grows faster than $1/\mu$, implying that $\mu\chi$ diverges as $\mu \to 0$. This behavior is consistent with the presence of chaos and absence of ergodicity. There is also an intermediate regime ($0.1 < x \lesssim 2$) in which the rescaled fidelity exhibits peculiar behavior. As we will discuss later, this regime has strong quantum finite $S$ effects which require a careful analysis.

In what follows, we will separately analyze the chaotic intermediate $x$ regime (Sec. 6.1), the small $x$ regime near integrability (Sec. 6.2), and confirm the absence of ergodicity throughout this model with other standard numerical probes (Sec. 6.4).

## 6.1 Far From Integrability

As we have discussed, the divergence of $\mu\chi$ as $\mu \to 0$ for $2 \lesssim x \lesssim 8$ indicates that the spectral function develops a low frequency tail, $\Phi_{ZZ}(\omega \to 0) \sim \omega^{-\gamma}$, where $\gamma$ is generally a nonuniversal exponent which can depend on $x$. This anticipated behavior agrees with numerical simulations (see solid line in Fig. 8(a)), where $\gamma(x = 4) \approx 0.625$. This scaling of the spectral function also agrees with the scaling of the fidelity $\chi \sim \mu^{-(1+\gamma)}$ shown in Fig. 8(b). In the classical limit these scalings are expected to last in the limit $\omega, \mu \to 0$, as there are no small parameters in the model which could introduce a low frequency scale, where this behavior would change.

The situation changes in the quantum finite $S$ regime. As the Hamiltonian (34) has a bandwidth of the order of unity and the Hilbert space dimension is $D \sim S^2$, the typical level spacing is on the order of $\Delta E \sim 1/D \sim 1/S^2$. This level spacing introduces a so-called Heisenberg scale $\omega_H \sim \Delta E/\hbar \sim 1/S$, which provides a natural low frequency cutoff for the spectral function. Surprisingly our numerical results, presented in Fig. 9, are better collapsed by the scale $\omega_S \sim 1/S^{3/4}$. As $S$ increases the spectral function approaches the ($S-$independent) classical asymptote over a broader range of frequencies. Using $\omega_S$ as the relevant low frequency scale, we anticipate that at large but finite $S$ the spectral function scales as

$$\Phi_{ZZ}(\omega) = \omega^{-\gamma} g_\Phi(\omega/\omega_S) = \omega_S^{-\gamma} \tilde{g}_\Phi(\omega/\omega_S) \quad \Rightarrow \quad \chi_{ZZ} = \frac{1}{\mu^{1+\gamma}} g_\chi(\mu/\omega_S) \qquad (52)$$

The corresponding scaling collapse is shown in the insets of Fig. 9 and works very well. We do not know if the exponent $3/4$, instead of the anticipated exponent 1, can be attributed to finite $S$ corrections or if there is another relevant quantum scale that is parametrically larger than the typical level spacing. Using this scaling ansatz we can extrapolate the spectral function to larger values of $S$ which cannot be accessed by exact diagonalization. These extrapolations are shown as dashed lines in Fig. 8.

We conclude this section by noting that the high frequency asymptotes of the spectral function shown in the inset of Fig. 8(a) are essentially indistinguishable from those of the integrable XXZ and XYZ models (Figs. 5 and 6, respectively). In each of these cases the spectral function decays exponentially, a feature which appears to be completely insensitive to whether the model is integrable or chaotic. We therefore

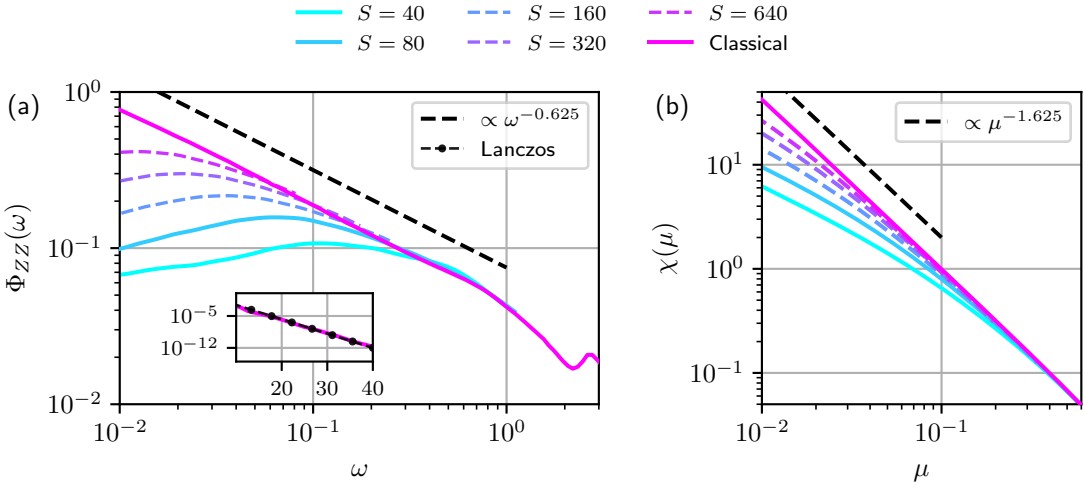

Figure 8: Spectral functions and fidelities of the chaotic model with $x = 4$. The classical data was computed by averaging $5 \times 10^5$ initial conditions simulated to a time $T = 6250$. The quantum data was computed by averaging 600 disorder realizations for $S = 40, 80$. Dashed lines indicate extrapolations to larger values of $S$ using the scaling ansatz (52). (a) In the classical limit, the low frequency spectral weight diverges as $\omega^{-0.625}$. Inset: the high-frequency regime exhibits exponential decay, with a decay rate $\tau \approx 1.158$ computed with Lanczos methods (nested Poisson brackets). (b) The corresponding fidelities, which obey the scaling ansatz (52).

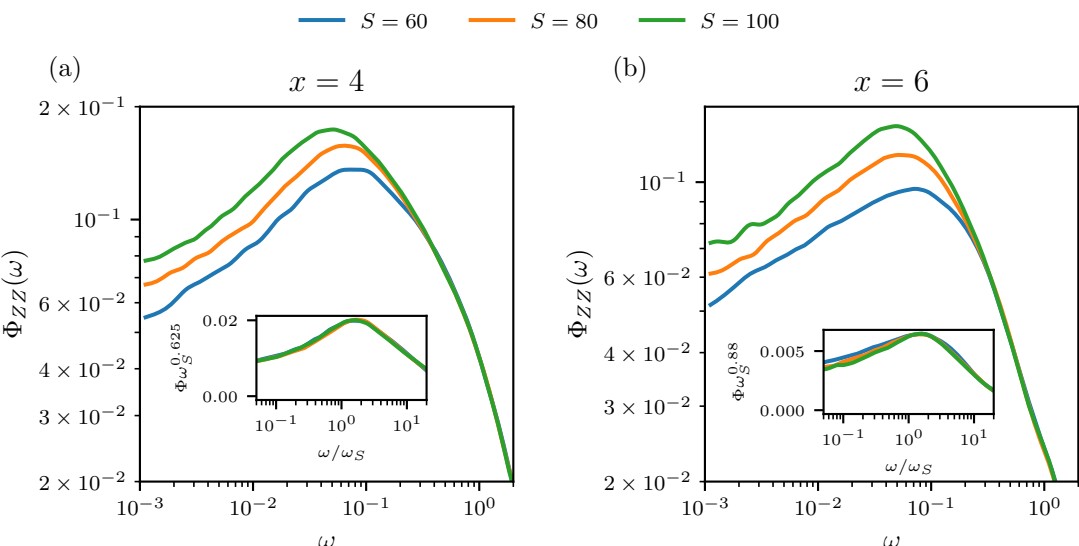

Figure 9: Spectral functions as a function of $S$ for (a) $x = 4$ and (b) $x = 6$. The insets show the best scaling collapse of $\Phi$ consistent with our data.

conclude that the short-time behavior of operators (equivalently, the growth of nested commutator norms, see App. B) cannot detect chaos in this system.

## 6.2 Near Integrability

As $x$ is reduced the system approaches the integrable point at $x = 0$ and we find very rich low-frequency behavior. Taking $x \sim 1$ as an example (see Fig. 10), we

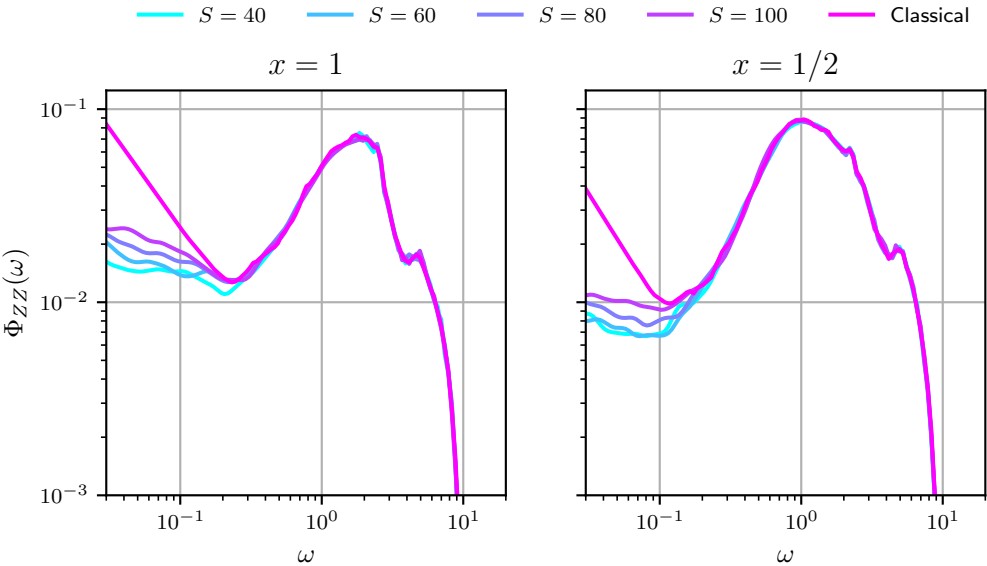

Figure 10: Spectral functions in the regime proximate to integrability for $x = 1$ (left) and $x = 1/2$ (right). Finite $S$ effects are particularly severe in this regime: the asymptotic low frequency properties of the classical spectral function converge very slowly as a function of $S$. It is instructive to compare this with the convergence highlighted in Fig. 8 (a).

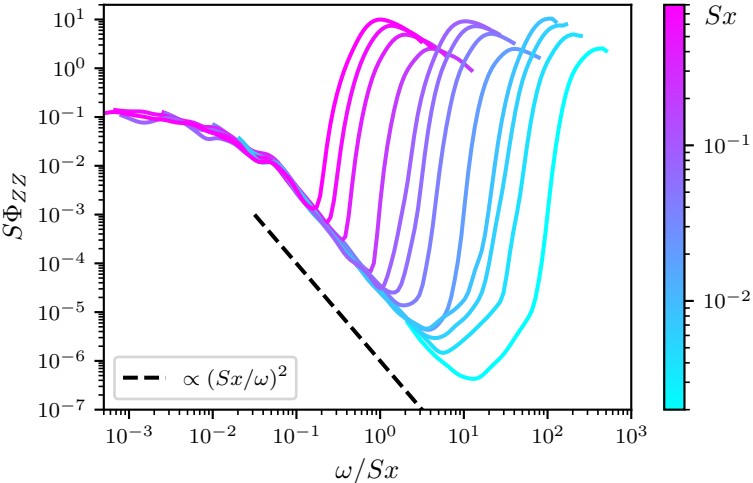

Figure 11: Quantum spectral functions in the small $x$ limit. The curves shown use the parameters $x = 10^{-2}, 10^{-3}, 10^{-4}$ and $S = 20, 40, 60, 80$, each averaged over several hundred disorder realizations. The collapse shown is consistent with perturbation theory in $x$.

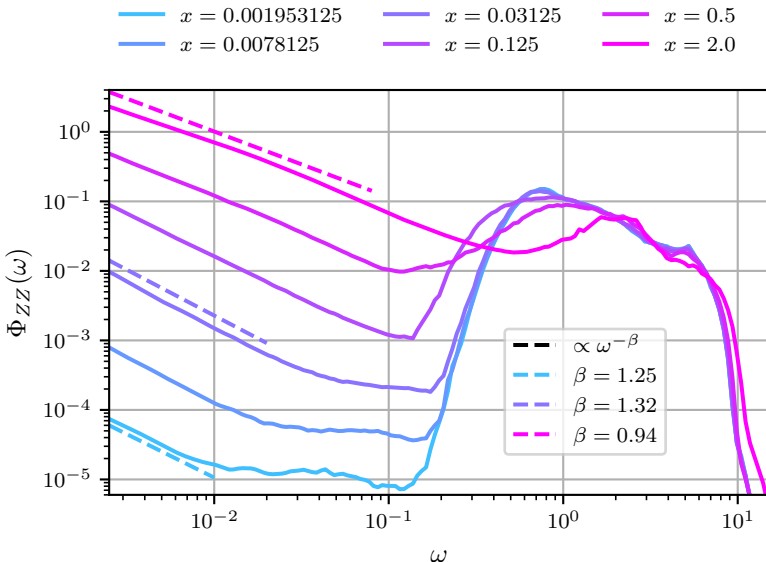

Figure 12: Classical spectral functions upon approaching the small $x$ limit. The low frequency spectral asymptotes are approximately parallel with similar power scalings $\Phi_{ZZ} \sim \omega^{-\beta}$ and non-universal values of $\beta$. Note that $\beta > 1$ cannot be an asymptotic result as it violates the sum rule $\int d\omega \Phi_{ZZ}(\omega) = \overline{ZZ(t=0)}^2_c < 1$.

observe that the spectral functions begin to decay as we lower the frequency from $\omega \sim 1$. This is precisely the behavior we expect, as the system has not had enough time to "realize" that it is chaotic. At sufficiently low frequencies (late times), the spectral weight reaches a minimum and begins to grow, indicating that the system is chaotic. Surprisingly, this growth is largely absent in quantum systems even when $\omega \gg \omega_H$, i.e., well above the Heisenberg scale. Since the classical limit is recovered by taking $S \to \infty$, we are forced to conclude that near integrability another quantum time scale develops which is much shorter than the Heisenberg time. Moreover, the reduced growth of spectral functions at finite $S$ implies that the fidelity diverges more slowly as a function of $\mu$, and in this sense the model becomes more chaotic with increasing $S$. This situation is somewhat reminiscent of the kicked rotor model, where due to Anderson localization the quantum kicked rotor is less chaotic than its classical counterpart [68]. Similar observations were made in the context of Arnold diffusion [69].

To proceed, we find it useful to focus on the limit $x \ll 1$ with finite $S$. In this regime one can use ordinary perturbation theory in $x$ to estimate the spectral function [41, 70, 71]:

$$\Phi_{ZZ}(\omega, S, x \ll 1) \sim CS \left(\frac{x}{\omega}\right)^2, \quad C \sim 3 \times 10^{-5} \tag{53}$$

The scaling $(x/\hbar\omega)^2$ is expected from standard perturbation theory; the additional factor of $1/S$ comes from the $1/\sqrt{S}$ suppression of the matrix elements of the integrability breaking perturbation $\partial_\lambda H$. Numerically, we have found an anomalously small overall prefactor $C \sim 3 \times 10^{-5}$, which depends neither on $S$ nor on $x$. This prefactor is likely of the same origin as the anomalously small prefactor in the $\omega^2$ asymptote of the XYZ model spectral function (see Fig. 6). The scaling (53) indeed agrees very well with the numerical results shown in Fig. 11.

The evolution of the spectral function at small $x$ as we approach the classical $S \to \infty$ limit is highly nontrivial. In this limit, like for larger $x$, we observe power-law scaling at small $\omega$: $\Phi_{ZZ}(\omega) \sim \omega^{-\beta}$ with $\beta \approx 1$ (see Fig. 12). Interestingly, the approximate $1/\omega$ dependence of the spectral function, which corresponds to logarithmically slow relaxation, is consistent with a similar behavior found in other systems

close to integrability, with and without disorder [34, 41]. The exponent $\beta = 1$ (up to logarithmic corrections) is the fastest asymptotic divergence of the spectral function consistent with the spectral theorem [34]. The fitted exponents slightly larger than one, which we found for some values of $x$, therefore cannot be asymptotic and are expected to decrease if we extend the analysis to even lower frequencies (and hence longer times). In turn, $\beta = 1$ corresponds to maximally fast asymptotic divergence of the fidelity susceptibility with the cutoff $\mu$ as illustrated in Fig. 1, meaning that long time trajectories are maximally unstable. Comparing this approximate scaling with the perturbative result (53), we see that one needs to reach $S \sim 1/C \sim 10^5$ even for reasonably large $x \approx 1$ in order to observe the crossover to the classical regime. The microscopic origin of this extremely large value of S remains mysterious and possibly related to strong quantum effects found in other systems close to integrability and, thereby, Arnold diffusion [69]. We note that this estimate is in line with the fact that Fig. 10 shows very slow convergence to the classical limit with increasing $S$.

### 6.3 Chaotic and Regular Regions of Phase Space

From the KAM theorem it is well known that in low-dimensional classical systems, like the one studied in this work, the phase space at small integrability breaking is mixed [2]. This means that some trajectories remain regular while some become chaotic. The spectral function and the fidelity susceptibility we studied so far were averaged over different initial conditions and thus can be regarded as average measures of chaos. In order to get additional information about the system, one needs to analyze their fluctuations as was done, e.g. in Refs. [34, 46, 72] for quantum systems. A detailed analysis of such fluctuations is beyond the scope of this work. However, we would like to illustrate that the spectral function (and thus the fidelity susceptibility) can be used to clearly distinguish regular and chaotic motion. This can be done by averaging over specific regions of phase space. Namely, we can analyze (43) and (44) where the phase space integrals only cover certain regions or initial conditions. Specifically, we consider spectral functions averaged separately over chaotic and regular regions, which show distinct low-frequency features.

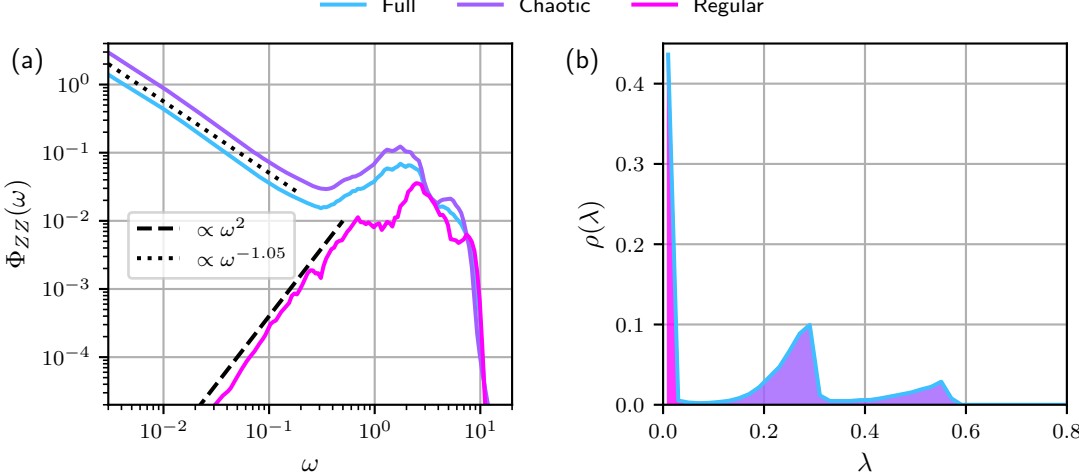

Figure 13: Classical spectral functions (left) and distribution of Lyapunov exponents (right) for chaotic and regular trajectories at $x = 1.5$. (a) shows distinct low-frequency behaviors of the spectral functions averaged over chaotic versus regular trajectories: asymptotes are $1/\omega^{1.05}$ and $\omega^2$, respectively. (b) showcases the probability distribution of the $5 \times 10^5$ trajectories considered for simulation time $T = 10^5$, with chaotic (purple) and regular (pink) ones separately colored.

We now focus on a specific value of $x = 1.5$. As one method for differentiating between chaotic and regular trajectories, we use the Lyapunov exponent $\lambda$, which describes the growth between two nearby trajectories $\mathbf{S}$ and $\widetilde{\mathbf{S}}$:

$$\lambda = \lim_{T \to \infty} \lim_{d(0) \to 0} \frac{1}{T} \log \left( \frac{d(T)}{d(0)} \right), \tag{54}$$

where $d(t) = ||\mathbf{S}(t) - \widetilde{\mathbf{S}}(t)||$ measures the distance between $\mathbf{S}$ and $\widetilde{\mathbf{S}}$. We use a method from Ref. [73] to numerically compute $\lambda$, where $d(0)$ and $T$ are finite and thus (54) sets the lower bound for the numerically accessible value of the Lyapunov exponent as $\lambda \gtrsim 1/T$. By examining the asymptotic behavior of $\lambda$ with respect to $T$, we can distinguish between regular trajectories with $\lambda \propto 1/T$ and chaotic trajectories with saturated values of $\lambda$ for sufficiently large $T$.

Using $\lambda$, we can compute the spectral functions for regular and chaotic trajectories, separately, and thus perform averaging over two different regions of phase space: Fig. 13(a) shows the spectral functions averaged over the entire, chaotic, and regular regions of phase space, while Fig. 13(b) shows the probability distribution $\rho(\lambda)$ of $\lambda$. We note that, due to numerical technicalities, the regular trajectories numerically have non-zero Lyapunov exponents in (b) that rather scale as $1/T$. However, $T$ is large enough to properly separate between regular and chaotic trajectories using the computed $\lambda$'s as indicated by a gap in the probability distribution between these two. We find distinct features of the low-frequency tails for the spectral functions averaged using chaotic versus regular trajectories. The former exhibits an approximate $1/\omega$ dependence that we have observed in Fig. 12, while the latter shows the expected $\omega^2$ scaling we saw for integrable models, such as in Fig. 6. Therefore, we find that chaos can be measured by observing either a non-zero Lyapunov exponent or non-decreasing low-frequency tail of the spectral function (or, equivalently, a non-decreasing scaling of $\mu\chi$ for $\mu \to 0$). Deriving precise mathematical connections between Lyapunov exponents and the low frequency asymptotes of spectral functions remains an unsolved problem.

## 6.4 Absence of Ergodicity

To complete our analysis of the chaotic model, we return to the question of whether or not it is ergodic at particular values of $x$. We remind the reader that, according to the eigenstate thermalization hypothesis [26], the spectral function should saturate below the Thouless frequency leading to $\chi \sim 1/\mu$ at small $\mu$. However, the analysis of sections 6.1 and 6.2 suggest that this is not the case in our chaotic model, particularly in the classical limit (see Figs. 8,12). To conclusively demonstrate that the model is not ergodic, in this section we compute other measures of ergodicity and confirm that they are consistent with the predictions of the fidelity susceptibilities and spectral functions.

A standard numerical test for ergodicity, motivated by random matrix theory, comes from the distribution of Hamiltonian eigenvalues. Given consecutive energy level spacings $s_n = E_{n+1} - E_n$, random matrix theory predicts that the probability distribution $P(s)$ is given by the Wigner-Dyson distribution, while integrable systems typically exhibit Poisson or nearly Poisson statistics. The computation of $P(s)$ for a particular Hamiltonian can require elaborate unfolding procedures; it is convenient to avoid these when possible by considering the distribution of $r$ values,

$$r_n = \frac{\min(s_n, s_{n+1})}{\max(s_n, s_{n+1})}. \tag{55}$$

which do not require unfolding the spectrum [74].

In Fig. 14, we present the average level spacing ratio $\langle r \rangle$ as a function of the integrability breaking parameter $x$, in addition to a representative level spacing distribution $P(s)$. The $r$-value never approaches the Wigner-Dyson result $\langle r_{WD} \rangle \approx 0.536$, although it does deviate significantly from the Poisson value $\langle r_P \rangle \approx 0.386$. Hence,

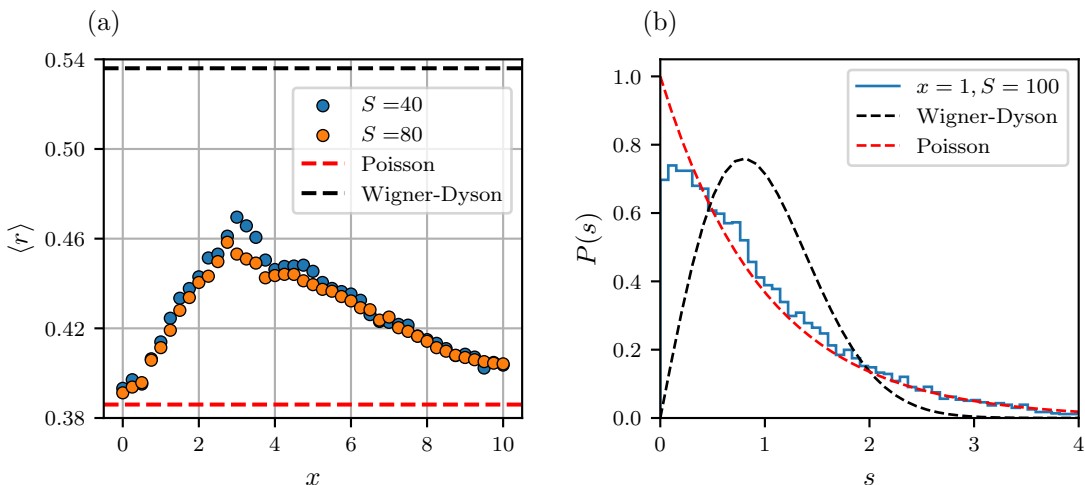

Figure 14: Energy-level spacing and corresponding $r$-statistics. (a) The average level spacing ratio $\langle r \rangle$ as a function of the integrability breaking parameter $x$ for different values of $S$, each averaged over 20 disorder realizations. (b) The level spacing distribution for $S = 100$, $x = 1$, keeping the central 50% of states in the spectrum. For this choice of parameters, density of states is approximately constant at the center of the spectrum and the level spacing distribution does not require a complicated unfolding procedure.

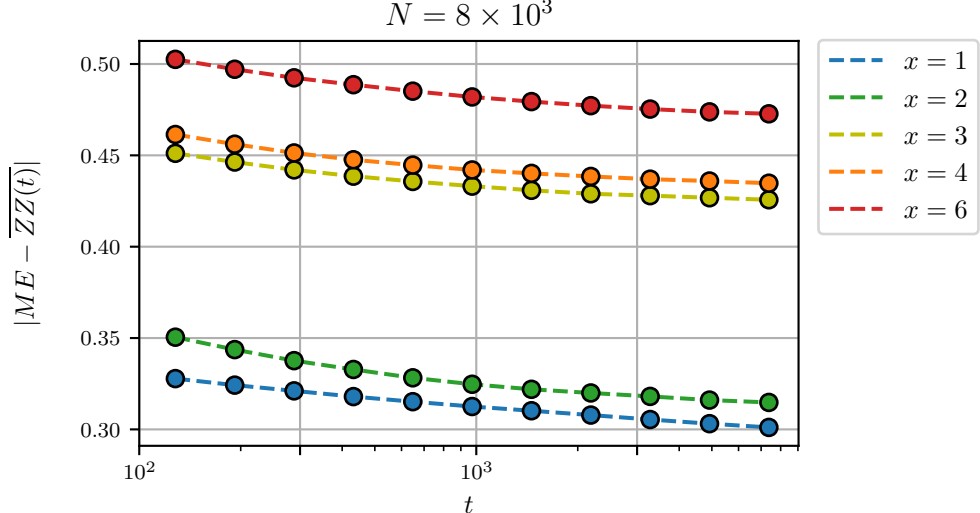

Figure 15: Statistical distance between the microcanonical ensemble and late-time expectation values in the classical limit. The microcanonical distribution was obtained by computing the expectation value of $ZZ$ in $10^6$ initial conditions within a narrow energy window at the center of the spectrum. The real-time data was computed by evolving $N = 8 \times 10^3$ random initial configurations in the same energy shell and computing the distribution of $ZZ(t)$. The data shown uses the $L_1$ distance between the microcanonical and real-time distributions.

while the chaotic model is clearly not integrable, it cannot be described by random matrix theory either, consistent with the presence of chaos and absence of ergodicity.

To check for the presence of ergodicity in the classical limit, we compare time-

averaged expectation values of the observable $ZZ(t)$ computed for individual trajectories with the corresponding predictions of the microcanonical ensemble. The resulting $L_1$ statistical distance is then averaged over initial conditions drawn from a microcanonical distribution. In Fig. 15, we compute the distance between the microcanonical ensemble and real-time expectation values averaged over a set of trajectories at fixed energy in the classical limit. Although we cannot rule out the possibility that the dynamics become effectively ergodic on timescales longer than those of our simulations, it is clear that the convergence to the microcanonical ensemble is extremely slow, if it happens at all. This is, again, consistent with the absence of ergodicity for all values of the integrability breaking parameter.

# 7    Conclusions

We have demonstrated that classical and quantum chaos can be probed and even defined based on the late time behavior of observables. Specifically, low frequency asymptotes of spectral functions quantify the sensitivity of quantum eigenstates and time averaged classical trajectories to small perturbations. This sensitivity is encoded in the scaling of the fidelity susceptibility (more generally, the quantum geometric tensor), which is equivalent to the norm of the generator of adiabatic transformations. We argued that unlike other definitions and probes of chaos, the properly regularized fidelity is well defined in both quantum and classical systems, regardless of the range of interactions or the size of the system, including the thermodynamic limit. Moreover, this probe allows one to clearly distinguish ergodic regimes from chaotic regimes which do not thermalize and separate regular and chaotic motion in the systems with mixed phase space. We confirmed that the qualitative phase diagram first proposed in Ref. [33] and schematically illustrated in Fig. 1 also applies to chaotic systems near the classical limit. In particular, we found that weakly nonintegrable systems first enter the maximally chaotic regime, which exhibits maximal growth of the fidelity susceptibility consistent with the spectral theorem. This regime is characterized by approximate $1/\omega$ scaling of the spectral function (also known as $1/f$-noise [75]) and corresponding very slow (logarithmic in time) relaxation of the system.

The arguments that we have presented to establish the fidelity as a probe of dynamics highlights the common structures underlying the emergence of chaos in both quantum and classical systems. Moreover, such an approach to studying the classical limit suggests a roadmap for how to detect different dynamical regimes in a broader range of systems than those studied in this work, including systems without a Hamiltonian description. We note that chaos defined in this way depends on the asymptotic scaling of the spectral function with the time (equivalently, frequency cutoff $\mu$). While at first glance such an approach might look unnatural, close to integrability many systems behave as if they are integrable for long times prior to exhibiting chaotic behavior.

Let us also comment on two other approaches to defining chaos which have recently been introduced, based on OTOCs [4,6] and operator spreading in Krylov space. [12]. Both approaches are based on the short time asymptotic behavior of observables. Interestingly, these probes seem to be valid in two opposing regimes. OTOCs are well-suited to classical or appropriate mean field limits, while Krylov methods apply to quantum systems in the thermodynamic limit. In this work, we found that, for systems close to the classical limit, the short-time behavior of operators, captured by the high frequency decay of spectral functions, cannot distinguish integrable and chaotic models. On the other hand, adiabatic transformations work equally well in both regimes. The question of how to connect these short- and long- time probes of dynamics remains an interesting and unsolved problem. Krylov space methods may offer a path for establishing such connections.

# 8 Acknowledgements

Some of the numerical computations in this work were performed using QuSpin [76,77]. The authors acknowledge helpful discussions with Anushya Chandran, Sumner Hearth, Tatsuhiko Ikeda, David Long, Gerhard Müeller, Dries Sels, Robin Schäfer, Gabe Schumm, and Dominik Vuina. H.K. and A.P. were supported by NSF Grant DMR-2103658 and the AFOSR Grant FA9550-21-1-0342. C.L. was supported by Boston University's REU program, which is supported by NSF Grant 1852266. K.M. is grateful to Leonid Levitov for organizing the Theoretical Physics Summer Practicum 2022, where this project was initiated. We also gratefully acknowledge technical support provided by Boston University's Research Computing Services.

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

# A  Spectral Asymptotes of the XXZ Model

This appendix analyzes the high and low frequency asymptotes of the spectral function $\Phi_{ZZ}(\omega)$ for the classical two-spin XXZ model of Sec. 4. In App. A.1, we derive closed-form expressions for the coordinate $z(t) = (S_1^z(t) - S_2^z(t))/2$ and the spectral function. We then analyze the asymptotic behavior of $\Phi_{ZZ}(\omega)$ at low frequencies (App. A.2) and high frequencies (App. A.3). Computation of the spectral function requires averaging over the set of initial conditions; in general, this average is difficult to perform analytically. When an analytic solution is not available, we are satisfied by deriving bounds on the asymptotes of $\Phi_{ZZ}(\omega)$. All special functions used in this section, particularly Jacobi functions and elliptic integrals, follow the conventions of Ref. [78].

## A.1  Solution by Quadratures

In this section, we solve for the trajectories $z(t)$ exactly and use them to write the spectral function as an average of closed-form expressions. As mentioned in the main text, $z(t)$ has been computed previously in Ref. [59]. Our approach emphasizes a different set of details to facilitate the computation of the spectral function and we present some additional results for localized trajectories.

In spherical coordinates, the spin components are given by

$$\boldsymbol{S}_i = (\sin\theta_i \cos\phi_i, \; \sin\theta_i \sin\phi_i, \; \cos\theta_i) \tag{56}$$

The equations of motion follow from the usual Hamiltonian treatment; we denote by $M_z$ and $E$ the conserved magnetization and energy of the XXZ Hamiltonian. Omitting tedious algebraic details, it is convenient to work with the coordinates

$$
\begin{aligned}
z &= \frac{1}{2}\left(S_1^z - S_2^z\right) = \frac{1}{2}\left(\cos\theta_1 - \cos\theta_2\right) \\
\xi &= \tan(\phi_1 - \phi_2)
\end{aligned}
\tag{57}
$$

The evolution of $z$ is completely decoupled from $\xi$:

$$
\begin{aligned}
\ddot{z} &= -2z\left(J_\perp^2 - J_z E + \left(J_\perp^2 - J_z^2\right)M_z^2\right) - 2z^3\left(J_z^2 - J_\perp^2\right) \\
&\equiv -V'(z).
\end{aligned}
\tag{58}
$$

Hence the function $z(t)$ can be interpreted as the position of a particle with unit mass in one dimension under the influence of a quartic potential $V(z) = \alpha z^2 + \beta z^4$, as claimed in the main text. This particle has an effective conserved energy $E_{\text{eff}}$ which is distinct from the Hamiltonian energy $E$:

$$
\begin{aligned}
E_{\text{eff}} &= \frac{1}{2}\dot{z}^2 + V(z) \\
&= \frac{1}{2}\left[J_\perp^2\left(1 - M_z^2\right)^2 - \left(E + J_z M_z^2\right)^2\right].
\end{aligned}
\tag{59}
$$

It is also useful to take note of the extremal points of the potential, defined by $V'(z_m) = 0$, and the particle's turning points, $z_0$, given by $E_{\text{eff}} = V(z_0)$,

$$
\begin{aligned}
z_m &= 0, \; \pm\sqrt{-\frac{\alpha}{2\beta}} \\
z_0^2 &= z_m^2 \pm \sqrt{z_m^4 + \frac{E_{\text{eff}}}{\beta}}
\end{aligned}
\tag{60}
$$

For some choices of parameters, a subset of these solutions may be complex; these are understood to be unphysical.

With these parameters in hand, rearrangement of Eq. (59) reduces the computation of $z(t)$ to quadratures. The trajectories then fall into two classes according to the sign of the particle energy $E_{\text{eff}}$, which we treat separately below. Some of the notation we employ in these cases is overloaded; this is useful for highlighting similarities between the solutions and it will always be clear from context which definitions should be used.

### A.1.1   $E_{\text{eff}} > 0$

Whenever $E_{\text{eff}} > 0$, the trajectory $z(t)$ is an even function about $z = 0$ and can be parameterized in terms of a new variable $\phi$ as $z = z_0 \sin \phi$:

$$
\begin{aligned}
t &= \pm \int_{z(0)}^{z(t)} \frac{dz}{\sqrt{2(E_{\text{eff}} - V)}} \\
&= \pm \frac{1}{\sqrt{2}} \int_{\sin^{-1}(z(0)/z_0)}^{\sin^{-1}(z(t)/z_0)} \frac{z_0 \cos \phi \, d\phi}{\sqrt{\alpha z_0^2 \cos^2 \phi + \beta z_0^4 \left(1 - \sin^4 \phi\right)}} \\
&= \pm \frac{z_0}{\sqrt{2E_{\text{eff}}}} \left[ F\left(\sin^{-1}\left(\frac{z(t)}{z_0}\right), -\frac{\beta z_0^4}{E_{\text{eff}}}\right) - F\left(\sin^{-1}\left(\frac{z(0)}{z_0}\right), -\frac{\beta z_0^4}{E_{\text{eff}}}\right) \right]
\end{aligned}
\tag{61}
$$

where $F$ is the incomplete Elliptic integral of the first kind. For the sake of notational clarity, we work with the positive branch of solutions and restore the negative branch later by enforcing TR symmetry explicitly. It is convenient to define $Q \equiv -\beta z_0^4 / E_{\text{eff}}$, and rearrangement of Eq. (61) yields

$$
\sin^{-1}\left(\frac{z(t)}{z_0}\right) = \text{am}\left[ \frac{\sqrt{2E_{\text{eff}}}}{z_0} t + \underbrace{F\left[\sin^{-1}\left(\frac{z(0)}{z_0}\right), Q\right]}_{\equiv \sqrt{2E_{\text{eff}}} t_0 / z_0}, Q \right]
\tag{62}
$$

$$
\Rightarrow z(t) = z_0 \text{sn}\left[ \frac{\sqrt{2E_{\text{eff}}}}{z_0} \left(t + t_0\right), Q \right]
$$

where sn is the Jacobi sine function and am is the Jacobi amplitude function. With $z(t)$ in hand, the spectral function $\Phi_{ZZ}(\omega)$ can be computed by averaging over initial conditions,

$$
\begin{aligned}
\Phi_{ZZ}(\omega) &= \frac{1}{2\pi} \overline{|ZZ(\omega)|^2} \\
&= \frac{1}{2\pi} \overline{\left[ \int dt \, e^{i\omega t} \frac{1}{2}\left(M_z^2 - z(t)^2\right) \right]^2}
\end{aligned}
\tag{63}
$$

Using the Fourier series representation of the elliptic functions, $ZZ(\omega)$ is given by

$$
\begin{aligned}
ZZ(\omega) &= \frac{\pi^2 z_0^2}{Q K(Q)^2} \sum_{n=1}^{\infty} \frac{n q^n}{1 - q^{2n}} \left[ e^{2i\omega_0 t_0} \delta(\omega + 2n\omega_0) + e^{-2i\omega_0 t_0} \delta(\omega - 2n\omega_0) \right] \\
&\quad + \frac{1}{2}\left( M_z^2 - \frac{z_0^2}{Q} + \frac{z_0^2 E(Q)}{Q K(Q)} \right) \delta(\omega)
\end{aligned}
\tag{64}
$$

where $q$ is the elliptic nome of $Q$, $K$ is the complete elliptic integral of the first kind, $E$ is the complete elliptic integral of the second kind, and $\omega_0 = \pi \sqrt{2E_{\text{eff}}}/(2K(Q)z_0)$ is the fundamental frequency of $z(t)$. Using Eq. (63), the spectral function is given by

$$
\begin{aligned}
\Phi_{ZZ}(\omega) &= \frac{1}{2\pi} \overline{\left( \frac{\pi^2 z_0^2}{Q K(Q)^2} \right)^2 \sum_{n=1}^{\infty} \left( \frac{n q^n}{1 - q^{2n}} \right)^2 \left[ \delta(\omega + 2n\omega_0) + \delta(\omega - 2n\omega_0) \right]} \\
&\quad + \frac{1}{8\pi} \overline{\left( M_z^2 - \frac{z_0^2}{Q} + \frac{z_0^2 E(Q)}{Q K(Q)} \right)^2} \delta(\omega).
\end{aligned}
\tag{65}
$$

### A.1.2   $E_{\text{eff}} < 0$

When $E_{\text{eff}} < 0$, it is necessarily the case that $\alpha < 0$ and $\beta > 0$, and all trajectories are localized in wells centered at $z_m = \pm\sqrt{-\alpha/2\beta}$. These trajectories are not symmetric

about $z = 0$ but $z^2(t)$ is symmetric under reflections about the center of the well. It is then useful to parameterize $z(t)^2 = z_m^2 + \delta z^2 \sin \phi$, where

$$\delta z^2 = \frac{2\sqrt{\alpha^2 + \beta E_{\text{eff}}}}{\beta} \tag{66}$$

Conservation of energy again reduces the computation of $z(t)$ to quadratures, and (restricting to the positive branch of solutions)

$$
\begin{aligned}
t &= \int_{z(0)}^{z(t)} \frac{dz}{\sqrt{2(E_{\text{eff}} - V)}} \\
&= \frac{\delta z^2}{2} \int_{\phi(z(0))}^{\phi(z(t))} d\phi \frac{\cos \phi}{2z\sqrt{E_{\text{eff}} - V}} \\
&= \frac{1}{z_m \sqrt{\beta}} \int_{\phi(z(0))}^{\phi(z(t))} \frac{d\phi}{\sqrt{1 + \frac{\delta z^2}{z_m^2} \sin \phi}}
\end{aligned}
\tag{67}
$$

For the sake of brevity, we define $a \equiv \delta z^2 / z_m^2$ and rearrange to obtain $z(t)^2$,

$$z(t)^2 = z_m^2 + \delta z^2 + 2\delta z^2 \text{sn}^2 \left[ \frac{z_m \sqrt{\beta(1+a)}}{2} (t + t_0), \frac{2a}{1+a} \right] \tag{68}$$

where $t_0$ is given by

$$t_0 = -\frac{2}{z_m \sqrt{\beta(1+a)}} F \left[ \frac{1}{4} \left( \pi - 2 \sin^{-1} \left( \frac{z(0)^2 - z_m^2}{\delta z^2} \right) \right), \frac{2a}{1+a} \right]. \tag{69}$$

It is convenient to define $Q \equiv 2a/(1+a)$, and just as in Sec. A.1.2, the spectral function is easily computed from the Fourier series representation of $z(t)^2$,

$$
\begin{aligned}
ZZ(\omega) = {}&\frac{2\pi^2 \delta z^2}{Q K(Q)^2} \sum_{n=1}^{\infty} \frac{n q^n}{1 - q^{2n}} \left[ e^{in\omega_0 t_0} \delta(\omega + n\omega_0) + e^{-in\omega_0 t_0} \delta(\omega - n\omega_0) \right] \\
&+ \left[ \frac{1}{2} \left( M_z^2 - z_m^2 - \delta z^2 \right) - \frac{\delta z^2}{Q} \left( 1 - \frac{E(Q)}{K(Q)} \right) \right] \delta(\omega)
\end{aligned}
\tag{70}
$$

where $q$ is the elliptic nome of $Q$ and $\omega_0 = \pi z_m \sqrt{\beta(1+a)}/(2K(Q))$ is the fundamental frequency of $z(t)$. Using Eq. (63), we find the spectral function

$$
\begin{aligned}
2\pi \Phi_{ZZ}(\omega) = {}&\overline{\left( \frac{2\pi^2 \delta z^2}{Q K(Q)^2} \right)^2 \sum_{n=1}^{\infty} \left( \frac{n q^n}{1 - q^{2n}} \right)^2 \left[ \delta(\omega + n\omega_0) + \delta(\omega - n\omega_0) \right]} \\
&+ \overline{\left[ \frac{1}{2} \left( M_z^2 - z_m^2 - \delta z^2 \right) - \frac{\delta z^2}{Q} \left( 1 - \frac{E(Q)}{K(Q)} \right) \right]^2} \delta(\omega).
\end{aligned}
\tag{71}
$$

## A.2   Low Frequency Asymptotes

In this section, we study the low-frequency behavior of $\Phi_{ZZ}(\omega)$. Rather than compute the asymptotic form of the spectral function exactly, we derive a set of bounds using a combination of the exact results of App. A.1 and numerics. As described in the main text, the qualitative results of this analysis depend on the sign of the quartic term $\beta$, and we treat the Heisenberg, easy-axis, and easy-plane regimes separately.

### A.2.1   The Heisenberg Point

At the Heisenberg point, $J_\perp = J_z \equiv J$ and the potential is purely quadratic, $V(z) = \alpha z^2$, $\alpha = J^2(1 + \boldsymbol{S}_1 \cdot \boldsymbol{S}_2)$. Each trajectory is harmonic and all of the properties of

the Heisenberg point follow from substituting $\beta = 0$ into the results of App. A.1.1. In particular, the spectral function follows from Eq. (65):

$$\Phi_{ZZ}(\omega) = \overline{\frac{(M_z^2 - z_0^2)^2}{8\pi}\delta(\omega) + \frac{z_0^4}{32\pi}\left[\delta(\omega - 2\omega_0) + \delta(\omega + 2\omega_0)\right]} \tag{72}$$

where $\omega_0 = \sqrt{2\alpha}$ is the fundamental frequency of $z(t)$ and

$$t_0 = \frac{1}{\omega_0}\sin^{-1}\left(\frac{z(0)}{z_0}\right). \tag{73}$$

By definition, $z_0 \leq 1$, so $\Phi_{ZZ}(\omega > 0)$ has a simple upper bound for each trajectory,

$$\Phi_{ZZ}(\omega > 0) \leq \overline{\frac{1}{32\pi}\delta(\omega - 2\omega_0)} \tag{74}$$
$$= \frac{P(\omega_0)}{32\pi}$$

where $P(\omega_0)$ is the probability of selecting an initial condition with fundamental frequency $\omega_0$. For initial conditions drawn from the infinite temperature distribution, it is clear that $\boldsymbol{S}_1 \cdot \boldsymbol{S}_2$ (and therefore $\alpha$) is uniformly distributed and it is straightforward to show that $P(\omega_0) = \omega_0/2J^2$ (see Fig. 16 (a)). Our upper bound for the spectral function at the Heisenberg point is therefore

$$\Phi_{ZZ}(\omega > 0) \lesssim \frac{\omega}{64\pi J^2}. \tag{75}$$

The numerics of Fig. 16 (b) indicate that this bound is quite tight.

Interestingly, the spectral function vanishes only linearly in the limit $\omega \to 0$, which is markedly distinct from the behavior in the easy-plane and easy-axis regimes. Of course, the Heisenberg point is unique due to its SU(2) symmetry and the perturbation $S_1^z S_2^z$ breaks this down to a U(1) symmetry. The consequences of this symmetry reduction at low energies is interesting, and developing a more complete understanding of the interplay between integrability, symmetry breaking, and the predictive properties of the fidelity susceptibility is an interesting avenue for future work.

### A.2.2  The Easy-Axis Regime

In the easy-axis regime, $J_\perp > J_z$ and the quartic term $\beta < 0$. Stability of the potential therefore requires that $\alpha > 0$; in fact, it is straightforward to show that

$$\alpha \geq J_\perp (J_\perp - J_z) \equiv \alpha_*. \tag{76}$$

Before going into mathematical details, let us recall the intuition suggested in the main text (see Fig. 4(a)). Low frequency trajectories correspond to either small oscillations about $z = 0$ with small $\alpha$ or trajectories which approach the maximum of the potential; since $\alpha$ is bounded from below, only the latter are relevant. For such trajectories, we expect that the period diverges as $T \sim \ln(|z_m - z_0|^{-1})$. We will now show how this intuition follows from our analytic results and use the constraints of the low frequency limit to bound $\Phi_{ZZ}(\omega)$.

For the easy-axis parameters, it is necessarily the case that $E_{\text{eff}} > 0$, so the results of App. A.1.1 apply to all trajectories. In this case, the fundamental frequency of a trajectory $z(t)$ is given by

$$\omega_0 = \frac{\pi\sqrt{2E_{\text{eff}}}}{2z_0 K(Q)} \tag{77}$$

Since $E_{\text{eff}} > 0$ and $z_0 \leq 1$, "small" frequencies can only appear if $K(Q)$ is large. Recall that $0 < Q = -\beta z_0^4/E_{\text{eff}} < 1$, and in that range $K(Q)$ is finite except in the limit $Q \to 1^-$. More precisely, for $Q = 1 - \epsilon, \epsilon \ll 1$,

$$K(Q) \sim \frac{1}{2}|\ln \epsilon| \tag{78}$$

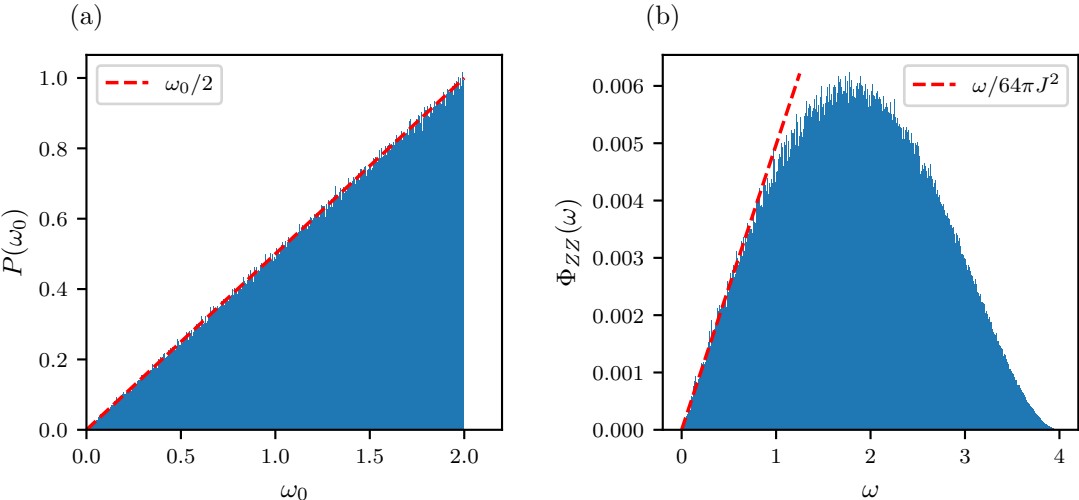

Figure 16: Spectral function at the Heisenberg point with $J = 1$. (a) A histogram of fundamental frequencies obtained by drawing $N = 10^6$ initial conditions randomly. This distribution converges to $\omega_0/2$. (b) The spectral function corresponding to the same initial conditions as in (a). The low frequency spectral weight scales linearly with $\omega$.

Trajectories with "small" $\omega_0$ therefore require exponentially precise fine-tuning of initial conditions, as expected. The meaning of a "small frequency" in this context is one which falls below the characteristic scale of the quadratic part of the potential, $\sqrt{2\alpha} \leq \sqrt{2\alpha_*}$ (see Eq. (76)).

Clearly, the low frequency limit is strongly constrained; numerics show that the probability of drawing an initial condition with fundamental frequency $\omega_0 \lesssim \sqrt{2\alpha_*}$ scales as

$$P(\omega_0 \lesssim \sqrt{2\alpha_*}) \sim \exp\left[-f(J_\perp, J_z)/\omega_0\right] \tag{79}$$

where $f$ is a function of the couplings alone (see Fig. 17 (a)). Due to the rapid decay of $P(\omega_0)$ as $\omega_0 \to 0$, the low frequency limit of the spectral function is dominated by the first harmonic of each trajectory. The spectral function of Eq. (65) is then approximated in the low frequency limit by

$$
\begin{aligned}
\Phi_{ZZ}(0 < \omega \lesssim \sqrt{2\alpha_*}) &\approx \frac{1}{2\pi} \overline{\left(\frac{\pi^2 z_0^2}{QK(Q)^2}\right)^2 \left(\frac{q}{1-q^2}\right)^2} \delta(\omega - 2\omega_0) + O\left(e^{-1/\omega}\right) \\
&\leq \frac{1}{2\pi} \left(\frac{\pi^2}{QK(Q)^2}\right)^2 \left(\frac{q}{1-q^2}\right)^2 \delta(\omega - 2\omega_0) + O(e^{-1/\omega}).
\end{aligned}
\tag{80}
$$

Each contribution to Eq. (80) is specified by two independent parameters of the corresponding trajectory, $Q$ and $\omega_0$. In the low frequency limit, these parameters are *not* independent: it is straightforward to show that $E_{\text{eff}} \to -2\beta$ and $z_0 \to 1$ as $\omega_0 \to 0$. From Eq. (77), it follows that, in the low frequency limit,

$$\omega_0 \approx \frac{\pi}{K(Q)} \sqrt{\frac{|\beta|}{2}}. \tag{81}$$

This relation reduces the bound of Eq. (80) to a single parameter. With this understanding, our final bound on the spectral function can be written as

$$\Phi_{ZZ}(0 < \omega \lesssim \sqrt{2\alpha_*}) \lesssim \frac{1}{2\pi} \left(\frac{\pi^2}{QK(Q)^2} \times \frac{q}{1-q^2}\right)^2 O(\exp\left[-1/\omega\right]). \tag{82}$$

The numerically-obtained spectral function is shown in Fig. 17 (b), and it is clear that the low frequency limit decays at least as fast as $\exp\left[-1/\omega\right]$.

(a)                                                                          (b)

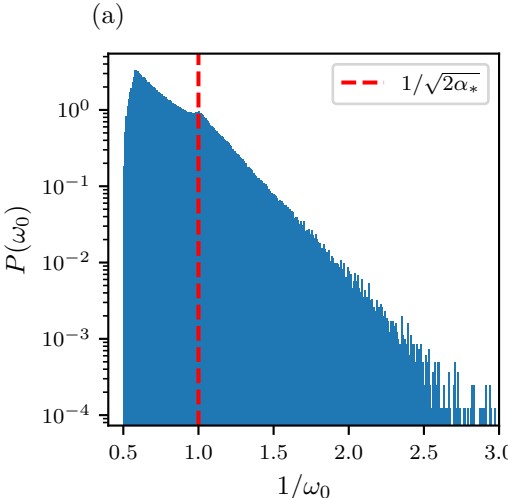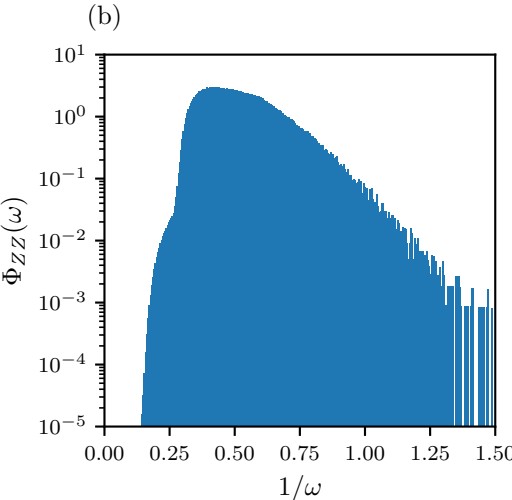

Figure 17: Spectral function of the easy-axis regime with $J_\perp = 1, J_z = 1/2$. (a) A histogram of fundamental frequencies obtained by drawing $N = 10^6$ initial conditions randomly. This distribution decays as $\exp\left[-1/\omega\right]$ at low frequencies. (b) The spectral function corresponding to the same initial conditions as in (a) as a normalized histogram, retaining the first ten harmonics. Clearly the spectral function decays faster than $P(\omega)$ at low frequencies; more precise bounds are discussed in the text.

### A.2.3    The Easy-Plane Regime

In the easy-plane regime, $J_\perp < J_z$ and the quartic term $\beta > 0$. Unlike the easy-axis case, stability considerations do not put a lower bound on $|\alpha|$ and $E_{\text{eff}}$ can be positive or negative. The mixture of signs in this case complicates the analysis, since some trajectories are subject to the results of App. A.1.1 and others to those of App. A.1.2.

Rather than focus on these complicated details, we use the intuition obtained from the easy-axis regime and focus on the distribution of fundamental frequencies. Numerics indicate that the probability of drawing a trajectory with fundamental frequency $\omega_0$ scales as

$$P(\omega_0) \sim \exp\left[-g(J_\perp, J_z)/\sqrt{\omega_0}\right]. \tag{83}$$

where $g$ is a function of the couplings alone (see Fig. 18 (a)). Since this distribution decays rapidly as $\omega_0 \to 0$, the spectral function is again dominated by the first harmonic of each trajectory. The spectral function in this case is then given by a weighted sum of contributions of the form shown in Eqs. (65) and (71). Each of these contributions are well-behaved functions, and we therefore expect that the exponential suppression of $P(\omega_0)$ sets a bound for the low-frequency behavior of the spectral function up to polynomial corrections in $\omega$. That is, despite the complicated form of the average trajectories, the leading behavior of $\Phi_{ZZ}(\omega \to 0)$ is set by $P(\omega_0)$. The numerics of Fig. 18 confirm this argument.

### A.3    High Frequency Decay Rates

In this section, we analyze the high frequency behavior of the spectral functions computed in App. A.1. We will argue that the spectral function decays exponentially in the high frequency limit,

$$\Phi_{ZZ}(\omega \to \infty) \sim \exp\left[-\tau_0 \omega\right] \tag{84}$$

where the decay rate $\tau_0$ is a function of the Hamiltonian couplings $J_\perp$ and $J_z$. In general, spectral decay rates can be determined using nested Poisson brackets or Lanczos

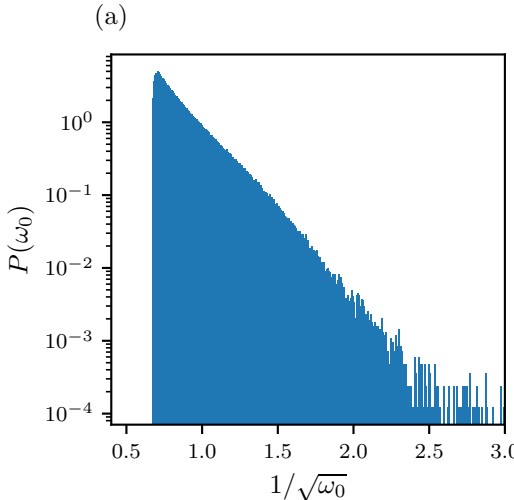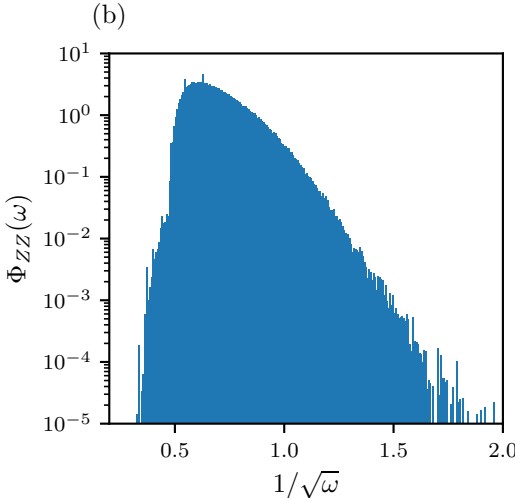

Figure 18: Spectral function of the easy-plane regime with $J_\perp = 1, J_z = 3/2$. (a) A histogram of fundamental frequencies obtained by drawing $N = 10^6$ initial conditions randomly. This distribution decays as $O(\exp[-1/\sqrt{\omega}])$ at low frequencies. (b) The spectral function corresponding to the same initial conditions as in (a) as a normalized histogram, retaining the first twenty harmonics. Clearly the spectral function decays faster than $P(\omega)$ at low frequencies.

methods (see App. B), but our analytic control over the XXZ model facilitates a more direct approach.

Take an arbitrary initial condition $\mathcal{S} = (\boldsymbol{S}_1, \boldsymbol{S}_2)$ with $E_{\text{eff}} > 0$; the generalization to $E_{\text{eff}} < 0$ will follow shortly. We denote the contribution of this initial condition to the spectral function by $\Phi_{ZZ}(\omega|\mathcal{S})$, which can be read off from Eq. (65). Dropping the Drude weight and irrelevant constants, we find

$$\Phi_{ZZ}(\omega > 0|\mathcal{S}) \propto \sum_{n=1}^{\infty} \left( \frac{nq^n}{1-q^{2n}} \right)^2 \delta\left(\omega - 2n\omega_0\right). \tag{85}$$

Note that the norm of $q$ is necessarily bounded, $|q| < 1$. At high frequencies, the amplitude of $\Phi_{ZZ}(\omega|\mathcal{S})$ decays as $q^{\omega/\omega_0}$, which gives a decay rate associated with $\mathcal{S}$:

$$\tau(\mathcal{S}) = -\frac{\ln|q|}{\omega_0} \tag{86}$$

Provided that the distribution of decay rates $\tau(\mathcal{S})$ has a well-defined minimum, we expect that the decay rate of the spectral function satisfies

$$\tau_0 = \min_{\mathcal{S}} \tau(\mathcal{S}) \tag{87}$$

Our numerics indicate that such a minimum exists for all the parameters that we have considered; a typical example of our findings is presented in Fig. 19.

For trajectories with $E_{\text{eff}} < 0$, the decay rate can be computed by the same argument and replacing the contribution $\Phi_{ZZ}(\omega|\mathcal{S})$ with the result Eq. (71). Again neglecting the Drude weight and negative constants, we find

$$\Phi_{ZZ}(\omega > 0|\mathcal{S}) \propto \sum_{n=1}^{\infty} \left( \frac{nq^n}{1-q^{2n}} \right)^2 \delta\left(\omega - n\omega_0\right) \qquad (E_{\text{eff}} < 0) \tag{88}$$

By comparing with Eq. (85), we see that the decay rate of a trajectory with $E_{\text{eff}} < 0$ is enhanced by a factor of two. This is because such trajectories are localized away from

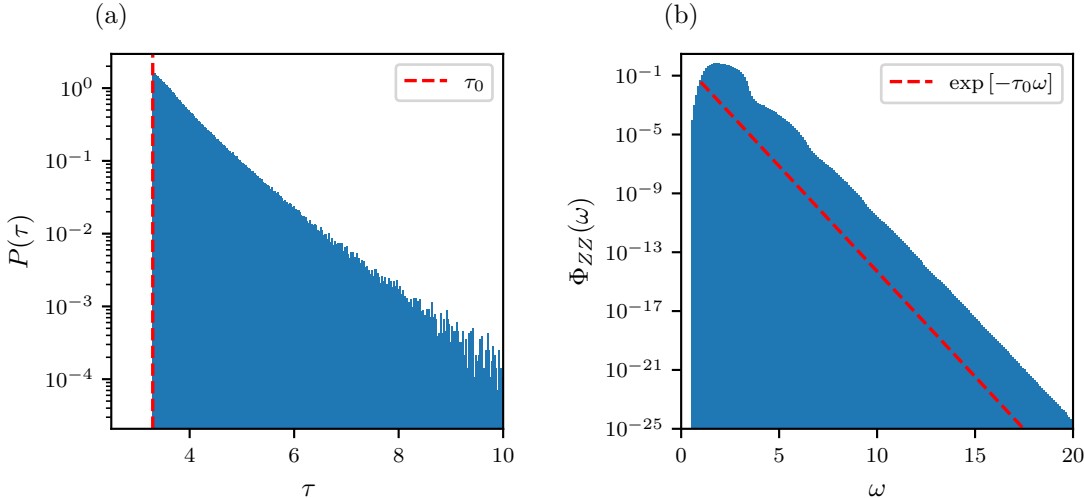

Figure 19: High-frequency decay rates for the XXZ model with $J_\perp = 1, J_z = 1/2$. (a) A histogram of decay rates obtained by drawing $N = 10^6$ initial conditions randomly. This distribution has a well-defined minimum $\tau_0$. (b) The spectral function corresponding to the same initial conditions as in (a) as a normalized histogram, retaining the first ten harmonics. The minimal decay rate $\tau_0$ clearly fits the high-frequency behavior of $\Phi_{ZZ}(\omega)$.

the origin, so the fundamental frequency of $z(t)$ is equal to that of $z(t)^2$; accordingly, each trajectory's amplitude decays as $q^{2\omega/\omega_0}$. In general, it is difficult to express the decay rates in a simpler form than Eq. (86). However, in cases where the quartic or quadratic parts of the potential $V(z)$ dominate, it is possible to make further analytic progress.

### A.3.1   $|\beta| \ll \alpha$

Here we study the high frequency decay rates of trajectories where the quadratic part of the potential $V(z)$ dominates the quartic part. We enforce this constraint by fixing $\beta$ and considering an initial condition with $|\beta| \ll \alpha$, which guarantees that $E_{\text{eff}} > 0$. Hence the decay rate $\tau$ of Eq. (86) is given by

$$\begin{aligned}
\tau &= \frac{\pi \, \text{Re}\,[K(1-Q)]}{\omega_0 K(Q)} \\
&= \frac{2z_0 \, \text{Re}\,[K(1-Q)]}{\sqrt{2E_{\text{eff}}}}
\end{aligned} \tag{89}$$

where we have used the definition of the elliptic nome and Eq. (77). At leading order in $\beta$, $z_0^2 \sim E_{\text{eff}}/\alpha$ and $Q = -\beta z_0^4/E_{\text{eff}}$ is a small negative number. Using known properties of elliptic functions, the leading asymptotic form of $\tau$ is given by

$$\tau \sim \sqrt{\frac{2}{\alpha}}\left[\ln 4 + \frac{1}{2}\ln\left(\frac{\alpha^2}{E_{\text{eff}}}\right) - \frac{1}{2}\ln|\beta|\right]. \tag{90}$$

By minimizing $\tau$ over the set of initial conditions, we obtain $\tau_0$; numerical results for $\tau_0$ are shown in Fig. 20, which shows that $\tau_0 \sim \log|\beta|$. In particular, at the Heisenberg point $\beta = 0$, the spectral function has a hard cutoff at $\omega = 2J$. Accordingly, our result predicts a logarithmic divergence of $\tau_0$ as $\beta \to 0$.

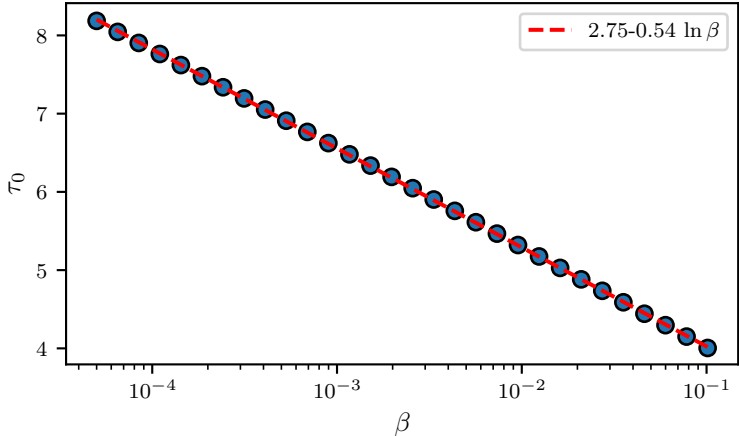

Figure 20: The spectral decay rate, $\tau_0$, obtained for $J_\perp = 1$ and varying $J_z$ to obtain the values of $\beta$ shown. The decay rate diverges logarithmically as $\beta \to 0$, consistent with the fact that the spectral function at the Heisenberg point has a hard cutoff at $\omega = 2J$. The fit line shown was obtained by a linear numerical fit.

### A.3.2   $\alpha = 0, \beta > 0$

In the absence of a quadratic term, the potential is entirely quartic, $E_{\text{eff}} = \beta z_0^4 > 0$ and $Q = -1$. Using Eq. (86), we find

$$
\begin{aligned}
\tau &= \sqrt{\frac{2}{E_{\text{eff}}/z_0^2}} \ \mathrm{Re}\left[K(2)\right] \\
&= \left(\frac{4}{\beta E_{\text{eff}}}\right)^{1/4} \mathrm{Re}\left[K(2)\right]
\end{aligned}
\tag{91}
$$

## B   Spectral Moments & Lanczos Methods

A number of previous works have explored the connection between operator growth and the well-known Lanczos algorithm [12, 79]. This appendix connects the contents of the main text to the Lanczos formalism and presents numerical results for both the spectral moments $R_n^2$ (see Eq. 32) and Lanczos coefficients. For the reader's convenience, we begin with a brief review of the Lanczos algorithm in the context of operator dynamics. We then apply these methods to our two-spin model, demonstrating consistency between spectral function decay rates and the growth rate of Lanczos coefficients.

### B.1   Brief Review of Lanczos Algorithm and Operator-State Formalism

For the sake of clarity, this section is written from the perspective of quantum mechanics and we use the terminology of finite-dimensional local Hilbert spaces. However, as explained in Sec. 2, all of our comments can be applied equally to classical Hamiltonian systems by replacing operators with suitable functions of phase space variables and substituting Poisson brackets for commutators (with appropriate factors of $i$ and $\hbar$).

In the Lanczos formalism, we begin with a Hilbert space $\mathcal{H}$, and it is convenient to regard operators which act on $\mathcal{H}$ as states in the "doubled" Hilbert space $\mathcal{H} \otimes \mathcal{H}$.

This is accomplished by the following mapping for an operator $\Omega$ acting on $\mathcal{H}$,

$$\Omega = \sum_{ij} \Omega_{ij} |i\rangle \langle j| \longrightarrow \sum_{ij} \Omega_{ij} |i\rangle \otimes |j\rangle \equiv |\Omega) \in \mathcal{H} \otimes \mathcal{H}. \tag{92}$$

In the last equality we have defined the "operator-state" corresponding to the operator $\Omega$, denoted by the rounded ket $|\Omega)$. The standard inner product on $\mathcal{H} \otimes \mathcal{H}$ is equivalent to the infinite temperature inner product for operators in the original Hilbert space, meaning that

$$(\chi|\Omega) = \frac{1}{\mathcal{D}} \operatorname{Tr} \chi^\dagger \Omega \tag{93}$$

where $\mathcal{D} = \dim(\mathcal{H})$. For systems described by a Hamiltonian $H$, time-evolution of a Hermitian operator $\Omega$ in the Heisenberg picture satisfies

$$\Omega(t) = e^{iHt/\hbar} \Omega e^{-iHt/\hbar} = \sum_{n=0}^{\infty} \frac{(it)^n}{n!} \mathcal{L}^n \Omega \tag{94}$$

where $\mathcal{L} = \hbar^{-1}[H, \cdot]$ is the Liouvillian superoperator. In the operator-state representation, the Liouvillian is an operator on the doubled Hilbert space given by

$$\mathcal{L} = \frac{1}{\hbar}\left(H \otimes \mathbb{1} - \mathbb{1} \otimes H^T\right) \tag{95}$$

and time evolution can be represented as $|\Omega(t)) = e^{i\mathcal{L}t}|\Omega)$. Note that the Liouvillian is Hermitian for real-symmetric Hamiltonians.

A nice tool for studying the dynamics of operators is provided by the Lanczos algorithm. In general, the Lanczos algorithm takes as input a Hermitian matrix $\Omega$ and a vector $\boldsymbol{v}$, and returns an orthonormal basis of vectors (the so-called Krylov basis) which tridiagonalizes $\Omega$. For our purposes, the Hermitian matrix we want to tridiagonalize is the Liouvillian and the initial vector is a (Hermitian) operator-state $|\mathcal{O}_0)$ whose evolution we wish to study. The Lanczos algorithm then proceeds as follows.

The operator $|\mathcal{O}_0)$ is the first element of the Krylov basis. The second element is constructed by noting that

$$(\mathcal{O}_0|\mathcal{L}|\mathcal{O}_0) \propto \operatorname{Tr}\left[\mathcal{O}_0\left(H\mathcal{O}_0 - \mathcal{O}_0 H\right)\right] = 0. \tag{96}$$

This motivates us to define $|A_1) = \mathcal{L}|\mathcal{O}_0)$, which we normalize to obtain the second Krylov basis element,

$$|\mathcal{O}_1) = \frac{|A_1)}{\sqrt{(A_1|A_1)}} \equiv |A_1)/b_1 \tag{97}$$

where we have also defined the first Lanczos coefficient, $b_1$.

The remainder of the algorithm can be described by induction. Let us assume that we have obtained the first $n$ Krylov basis vectors which satisfy $(\mathcal{O}_i|\mathcal{O}_j) = \delta_{ij}$, along with the first $n-1$ Lanczos coefficients. Define

$$|A_n) = \mathcal{L}|\mathcal{O}_{n-1}) - b_{n-1}|\mathcal{O}_{n-2}) \tag{98}$$

along with the $n$th Lanczos coefficient, $b_n = \sqrt{(A_n|A_n)}$. We will now show that $|A_n)$ is orthogonal to the first $n$ Krylov basis vectors. A calculation which is essentially identical to Eq. (96) shows that $(\mathcal{O}_{n-1}|A_n) = 0$. For all $m \leq n-2$, consider the overlaps

$$\begin{aligned}
(\mathcal{O}_m|A_n) &= (\mathcal{O}_m|\mathcal{L}\mathcal{O}_{n-1} - b_{n-1}\mathcal{O}_{n-2}) \\
&= (\mathcal{L}\mathcal{O}_m|\mathcal{O}_{n-1}) - b_{n-1}\delta_{m,n-2} \\
&= (A_{m+1} + b_m\mathcal{O}_{m-1}|\mathcal{O}_{n-1}) - b_{n-1}\delta_{m,n-2} \\
&= (A_{m+1}|\mathcal{O}_{n-1}) - b_{n-1}\delta_{m,n-2} \\
&= 0.
\end{aligned} \tag{99}$$

Element $n + 1$ of the Krylov basis is therefore given by the normalized vector

$$|\mathcal{O}_n) = |A_n)/b_n. \tag{100}$$

The Krylov basis is then constructed by iteration until one finds $b_n = 0$ for some $n$, at which point the basis has been found[4].

The relation (98) guarantees that the Liouvillian is a symmetric tridiagonal matrix in the Krylov basis, with zeros on the diagonal:

$$(\mathcal{O}_n|\mathcal{L}|\mathcal{O}_m) = \begin{pmatrix} 0 & b_1 & 0 & 0 & \cdots \\ b_1 & 0 & b_2 & 0 & \cdots \\ 0 & b_2 & 0 & b_3 & \cdots \\ \vdots & \vdots & \ddots & \ddots & \ddots \end{pmatrix}. \tag{101}$$

We note that the standard treatment of the Lanczos algorithm is slightly different than the one presented here, as it is in general possible to have nonzero diagonal entries following the Lanczos procedure.

The Krylov basis has several convenient features - for example, it reduces the problem of operator dynamics to the solution of a tight binding model whose hopping amplitudes are given by the Lanczos coefficients. Unfortunately, the Lanczos algorithm is susceptible to strong numerical instabilities, which can limit its application to many-body problems. However it has recently been emphasized, particularly in Ref. [12], that the Lanczos coefficients contain important universal information.

## B.2    Numerical Results

Using the operator-state formalism discussed in the previous section, we can relate dynamical properties of an operator to the Lanczos coefficients directly. For example, the autocorrelation function of a Hermitian operator $\partial_\lambda H$, denoted $C(t)$, can be written as

$$C(t) = (\partial_\lambda H|e^{i\mathcal{L}t}|\partial_\lambda H) = \frac{1}{\mathcal{D}} \operatorname{Tr}\left[\partial_\lambda H(t)\partial_\lambda H(0)\right]. \tag{102}$$

The associated spectral function, $\Phi(\omega)$, is the Fourier transform of the autocorrelation function. Moments of the spectral function are given by

$$R_n^2 = \int d\omega \, \omega^{2n} \Phi(\omega) = (\partial_\lambda H|\mathcal{L}^{2n}|\partial_\lambda H) = ||\mathcal{L}^n \partial_\lambda H||^2. \tag{103}$$

The attentive reader will notice that this form is equivalent to that in Eq. (31) in the limit of infinite temperature. One can easily check using Eq. (101) that $R_1 = b_1^2, R_2 = b_1^4 + b_1^2 b_2^2$, and so on.

The asymptotic growth of the Lanczos coefficients controls the high-frequency behavior of the spectral function,

$$b_n \sim n^\delta \Leftrightarrow \Phi(\omega \to \infty) \sim \exp\left[-(\omega/\omega_0)^{1/\delta}\right]. \tag{104}$$

where $\delta$ is a positive real number and $\omega_0$ is a frequency scale.

For local Hamiltonian systems, Ref. [12] has argued that $\delta \leq 1$ and, in the case $\delta = 1$, refined these asymptotic bounds by using the growth rate of the Lanczos coefficients, $\alpha$,

$$b_n \sim \alpha n \Leftrightarrow \Phi(\omega \to \infty) \sim \exp\left[-\frac{\pi\omega}{2\alpha}\right]. \tag{105}$$

Fig. 21 shows the first twenty Lanczos coefficients and spectral moments for the chaotic model of Sec. 6 with $x = 4$. We have taken special care to guarantee the numerical stability of both sets of quantities; for finite values of $S$, the Lanczos coefficients and spectral moments were computed exactly by constructing a finite-dimensional

---

[4]We note that the Krylov basis is not necessarily complete; for example, in the presence of symmetries, different symmetry sectors are dynamically disconnected.

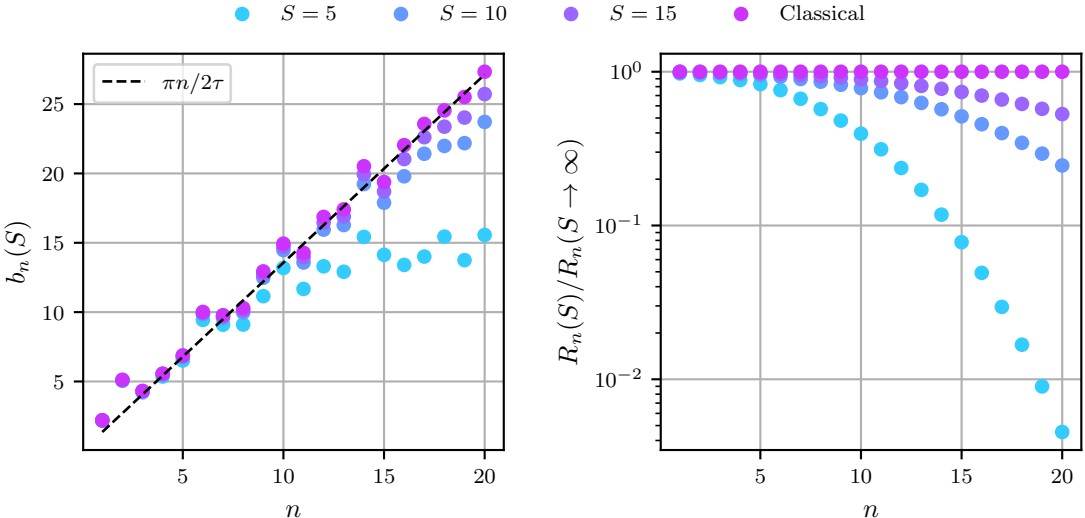

Figure 21: Lanczos coefficients and spectral moments of the chaotic two-spin model of Sec. 6 with $x = 4$. (a) The Lanczos coefficients associated with the operator $S_1^z S_2^z$ at several values of $S$. Finite $S$ effects lead to eventual saturation and decay of the Lanczos coefficients, analogous to finite size effects in quantum spin chains. In contrast, the classical coefficients can grow without bound. The black dashed line shows the growth rate anticipated by the high frequency decay of the spectral function, $\tau \approx 1.158$ (see Fig. 8). (b) The spectral moments associated with the Lanczos coefficients of panel (a), normalized by the classical data. This normalization reveals that seemingly small variations of the Lanczos coefficients are amplified dramatically in the moments.

representation of the Liouvillian. In the classical limit, we used matrix-free methods discussed in Ref. [12], which can be applied to systems with a countably infinite Hilbert space dimension, although they are fundamentally limited by computer memory. These approaches guarantee numerical stability, while other methods that rely on exact diagonalization are plagued by numerical instabilities.

Both the moments and Lanczos coefficients converge to their classical values in the limit $S \to \infty$, although the convergence is subtle. In addition to finite size effects which come from the fact that the space of operators has a finite dimension $(2S + 1)^4$, the Hamiltonian itself depends explicitly on $S$, meaning that the Lanczos coefficients $b_n(S)$ can disagree even at small $n$. This is to be contrasted with taking the thermodynamic limit at fixed $S$, for which Lanczos coefficients are guaranteed to agree exactly prior to encountering finite-size effects.