# Peer review of "Defining classical and quantum chaos through adiabatic transformations"

_SciPost Physics_

## Round 1 · Referee Report · Tigran Sedrakyan (Referee 1) · 2024-7-20

Strengths
1. The manuscript tackles an important and timely topic in theoretical physics, providing a unified approach to understanding chaos in both classical and quantum domains.
2. The use of fidelity susceptibility as a measure of chaos is innovative and offers new insights into distinguishing between integrable, chaotic but non-thermalizing, and ergodic regimes.
3. The paper is well-organized, and the arguments are generally clear and supported by thorough theoretical and numerical analysis.
Weaknesses
1. While the manuscript provides a comprehensive overview of the fidelity susceptibility and its relation to chaos, the connection between this measure and traditional notions of chaos (e.g., Lyapunov exponents) needs further elaboration. A more detailed discussion on how the proposed formalism aligns or diverges from classical chaos indicators would be beneficial.
2. The manuscript mentions the slow convergence of quantum spectral functions to their classical counterparts near integrability. To highlight this convergence behavior explicitly, it would be useful to include more quantitative comparisons between quantum and classical results, possibly through additional figures or detailed discussions. Perhaps more details on the numerical methods should be provided, especially for the exact diagonalization and the treatment of finite-size effects.
3. Identifying and characterizing the intermediate regime between integrable and ergodic behavior is intriguing. Additional examples or case studies illustrating the intermediate regime would enhance the reader's understanding.
Report
The manuscript addresses a significant and complex topic with a novel approach that has the potential to advance our understanding of chaos in classical and quantum systems. Given that the authors address the points raised in this report, I would recommend the manuscript for publication in SciPost Physics.
Recommendation
Publish (easily meets expectations and criteria for this Journal; among top 50%)
Author: Michael Flynn on 2024-12-17 [id 5046]
(in reply to Report 1 by Tigran Sedrakyan on 2024-07-20)
We thank the referee for his attentive reading of our manuscript and for his suggested revisions/questions. We will respond to the weaknesses he identified below:
>While the manuscript provides a comprehensive overview of the fidelity susceptibility and its relation to chaos, the connection between this measure and traditional notions of chaos (e.g., Lyapunov exponents) needs further elaboration. A more detailed discussion on how the proposed formalism aligns or diverges from classical chaos indicators would be beneficial.
We have added significant pedagogical discussion on these points to the introduction (Sec. 1), including an example which illustrates the fidelity susceptibility approach to chaos (see Fig. 2 and surrounding discussion). These discussions address the referee's concerns from a number of perspectives.
First, we point out that measures such as Lyapunov exponents have some conceptual inconsistencies: in particular, the existence of large Lyapunov exponents does not imply the existence of long-time and long-distance instabilities (``the butterfly effect''). This is to say nothing of the obvious difficulty of interpreting Lyapunov exponents in generic quantum mechanical systems. Second, the example treated in Fig. 2 provides an intuitive approach to diagnosing chaos that is equally applicable to both quantum and classical systems, circumventing the issues raised with Lyapunov exponents.
Beyond these high-level introductory points, we have performed additional calculations for the two-spin model studied throughout the remainder of the text. In particular, Sec. 6.3 of the revised manuscript introduces the concept of partial phase-space averaging: here, the idea is to separate trajectories into regular and chaotic domains. In Fig. 13 (b), we categorize trajectories into regular and chaotic domains based on their Lyapunov exponents, where regular trajectories are defined to have exponents which tend towards zero in the limit of long simulation times. Using this criteria, we also compute the spectral functions of trajectories corresponding to chaotic and regular regions (Fig. 13 (a)). The results of this analysis reveal that the fidelity susceptibility can be used to analyze chaos in different regimes of phase space; in particular, the presence of a mixed phase space or KAM region is not an obstacle to the application of our method. In combination with existing dynamical probes, our approach has the ability to separate chaos and ergodicity as dynamical concepts, as discussed throughout the manuscript.
>The manuscript mentions the slow convergence of quantum spectral functions to their classical counterparts near integrability. To highlight this convergence behavior explicitly, it would be useful to include more quantitative comparisons between quantum and classical results, possibly through additional figures or detailed discussions. Perhaps more details on the numerical methods should be provided, especially for the exact diagonalization and the treatment of finite-size effects.
Regarding the issue of numerical details, we refer the referee to Sec. 3.1, which provides an overview of our numerical methods in general. When specific calculations require us to deviate from the methods outlined in Sec. 3.1, we provide a complete discussion of our methods in the relevant section.
On the subject of convergence between quantum and classical data near integrability, we have significantly expanded the material in Sec. 6.2. In particular, we have performed additional calculations (see Fig. 12) which examine classical spectral functions as the integrability breaking parameter, $x$, is reduced. We find that as $x$ becomes small the classical results for the spectral function at low frequencies have very slow convergence to the asymptotic integrable limit. Similarly, at fixed small $x$ the low-frequency tail of the spectral function does not reach asymptotic behavior within the range of numerically accessible frequencies, indicating extremely slow convergence. We discuss the implications of these results in relation to our quantum data and estimate that convergence requires $S\sim 10^5$ even when $x$ is of order $1$. Conceptually, our results are qualitatively similar to observations in the context of Arnold diffusion for which we have added a corresponding reference.
>Identifying and characterizing the intermediate regime between integrable and ergodic behavior is intriguing. Additional examples or case studies illustrating the intermediate regime would enhance the reader's understanding.
We agree that this is an intriguing regime, and a number of works have pioneered efforts to explore it (see for example Refs. [33, 41] of our revised manuscript). It seems that the phenomena which can appear in this regime are quite diverse; in the case of our model, we see anomalously long relaxation times in the classical limit and anomalously slow recovery of classical data from quantum data. In addition, we have argued that our model does not exhibit thermalization for any choice of couplings (see Sec. 6.4). Further examinations of this regime are beyond the scope of the current work, but we hope the revised discussions of the introductory section will help sate the reader's curiosity.
Author: Michael Flynn on 2024-12-17 [id 5047]
(in reply to Report 2 on 2024-09-05)We thank the referee for their careful reading of our manuscript and for their detailed questions. Below are our responses:
Per the referee's suggestion, we have added explicit labels to figures 3 and 4 which should address their concerns. In the case of Fig. 1, that figure is meant to be understood as a sketch, but we agree that our initial presentation of the figure was too sketchy. We have added clarifying comments which should address the referee's concerns.
This concern is similar to others raised in the first referee report; as such, we refer the referee to our answer there for more complete details. To briefly recapitulate, we have expanded the discussions of the introduction (Sec. 1) to better reflect the relationship between our probe, the fidelity susceptibility, and other well-known dynamical probes, particularly Lyapunov exponents. In addition, the new section 6.3 shows how Lyapunov exponents can be used to separate trajectories into regular and chaotic regions in a mixed phase space, revealing new features of the fidelity susceptibility. We believe that these Lyapunov exponents provide similar information to Poincaré maps while being somewhat more quantitative.
We have partially addressed these comments already, but we would like to return to the question of which results are novel in this work. We agree with the referee that our primary contribution is to establish the utility of the fidelity susceptibility, which is expressed through the long time response as a probe of chaos and ergodicity in classical systems (prior work has already shown this for quantum systems). However, it is also important to acknowledge that our conceptual understanding of classical chaos is rather different from (although wholly consistent with) the existing literature. In particular, our choice to focus on the fidelity susceptibility offers a unified perspective on both quantum and classical chaos and ergodicity. The standard classical perspective focused on Lyapunov exponents runs into severe difficulties as exemplified in our new discussion.
For example, we know that weakly non-integrable systems with small Lyapunov exponents (such as air) which allow turbulence are much more chaotic than strongly nonintegrable systems such as water. We also expanded Fig. 2 to highlight that the definition of chaos based on fidelity is extremely intuitive as it targets the long-time instability of time averaged probability distributions (quantum or classical) to small perturbations. We do not understand the mathematical connections (if any) between Lyapunov exponents and long-time instabilities. Likewise, we do not understand connections (if any) between short-time operator growth, recently conjectured to be a measure of quantum chaos, and long-time response. These remain very interesting unsolved questions. By now, we have several additional many-particle classical models, such as the central spin model and a perturbed Ishimori chain, where we see again that the fidelity approach to chaos works very well. Our colleagues at BU in the group of D. Campbell applied this approach to contrast the FPUT chain versus the integrable Toda lattice, and again results very convincingly show that the FPUT model is chaotic in the long time limit. Those results will be reported in separate works.
We thank the referee for raising this important point. The main reason our response took so long is that we wanted to investigate this issue very carefully. The results are actually in line with our message and agree with the points raised by the referee. In response to these comments, we added the whole new section 6.3, updated Fig. 2, and added some relevant comments throughout the text.
Our work focuses primarily on the phase space averaged fidelity susceptibility, which is analogous to the phase space averaged Lyapunov exponent. Such measures do not see inhomogeneities in phase space. We believe it is very interesting to address the question of fluctuations of the fidelity susceptibility, which must be very strong in the mixed phase space regime. In quantum disordered systems these questions were addressed in Ref. https://arxiv.org/pdf/2009.04501 (see Fig. 6), where, in a similar chaotic but non-ergodic regime, the fluctuations of the fidelity susceptibility are very large. This is also consistent with what the referee wrote about quantum systems. To partially address this point in the current work, we analyzed the behavior of the spectral function and fidelity susceptibility in a regular part of the phase space and found that according to our criteria it is integrable, while in the chaotic part of phase space we found that the system is chaotic, as expected. When we do full phase space averaging of the fidelity susceptibility, the results are dominated by the chaotic phase space, which leads to a divergent response. In this respect, our approach to defining chaos through the fidelity susceptibility does not contradict the existing literature.

---

## Editorial Decision

unknown